# DANCING IN CHAINS: STRATEGIC PERSUASION IN ACADEMIC REBUTTAL VIA THEORY OF MIND

**Zhitao He** [†]**, Zongwei Lyu** [†]**, Yi R. (May) Fung** [✉]
Hong Kong University of Science and Technology
MMSense Lab
zhebu@cse.ust.hk, yrfung@ust.hk

## ABSTRACT

Although artificial intelligence (AI) has become deeply integrated into various stages of the research workflow and achieved remarkable advancements, academic rebuttal remains a significant and underexplored challenge. This is because rebuttal is a complex process of strategic communication under severe information asymmetry rather than a simple technical debate. Consequently, current approaches struggle as they largely imitate surface-level linguistics, missing the essential element of perspective-taking required for effective persuasion. In this paper, we introduce **RebuttalAgent**, the first framework to ground academic rebuttal in Theory of Mind (ToM), operationalized through a ToM-Strategy-Response (TSR) framework that models reviewer mental state, formulates persuasion strategy, and generates evidence-based response. To train our agent, we construct **RebuttalBench**, a large-scale dataset synthesized via a novel critique-and-refine approach. Our training process consists of two stages, beginning with a supervised fine-tuning phase to equip the agent with ToM-based analysis and strategic planning capabilities, followed by a reinforcement learning phase leveraging the self-reward mechanism for scalable self-improvement. For reliable and efficient automated evaluation, we further develop **Rebuttal-RM**, a specialized evaluator trained on over 100K samples of multi-source rebuttal data, which achieves scoring consistency with human preferences surpassing powerful judge GPT-4.1. Extensive experiments show RebuttalAgent significantly outperforms the base model by an average of 18.3% on automated metrics, while also outperforming advanced proprietary models across both automated and human evaluations. *Disclaimer: the generated rebuttal content is for reference only to inspire authors and assist in drafting. It is not intended to replace the author's own critical analysis and response.*[1]

## 1 INTRODUCTION

Large language models (LLMs) are profoundly reshaping the entire research workflow (Liu et al., 2024b; Lu et al., 2024; Chen et al., 2025b), from acting as a powerful tool for auxiliary tasks such as literature summarization (El-Kassas et al., 2021; Koh et al., 2022) and data visualization (Waskom, 2021; Wu et al., 2021), to serving as a collaborative partner in core tasks such as hypothesis formulation (Wang et al., 2024; Novikov et al., 2025; He et al., 2025b) and experimental design (Wang et al., 2021; Huang et al., 2024), and even functioning as an autonomous author of complete scientific papers that successfully pass human peer review (Weng et al., 2024; Schmidgall & Moor, 2025). While LLMs have become an indispensable collaborator in most stages of research, its role in the critical phase of *academic rebuttal* remains underexplored. From a game-theoretic perspective, the academic rebuttal process is not a simple technical debate but rather a classic Dynamic Game of Incomplete Information (Başar & Olsder, 1998; Fudenberg & Tirole, 1991; Owen, 2013). In this process, authors must persuade reviewers under severe *information asymmetry*, whereby they are unaware of the reviewers' knowledge base, intrinsic biases, or the cascading effects of their responses.

---

[✉] Corresponding author.
[†] Equal contribution
[1] Our code and models: https://github.com/Zhitao-He/RebuttalAgent

Current approaches for addressing this challenge, which primarily rely on Supervised Fine-tuning (SFT) on review datasets (Zhang et al., 2025), suffer from the fundamental limitations of direct imitation. These models excel at mimicking surface-level linguistic patterns, resulting in responses that are superficially polite but often formulaic and lack strategic depth. This failure stems from their inability to perform the strategic, perspective-taking reasoning demanded by the game-theoretic structure of rebuttal. In practice, a successful rebuttal transcends superficial politeness and is, at its core, an exercise in strategic reasoning (Harland et al., 2017; Palminteri, 2023; Lim & Bowman, 2024). This requires a complex analysis of trade-offs, such as when to concede, when to stand firm, when to provide new evidence, or when to reframe the narrative. Successfully navigating these trade-offs depends on the ability to perceive the mind of the other, a capacity known in cognitive science as **_Theory of Mind (ToM)_** (Wellman, 2002; Leslie et al., 2004; Goldman et al., 2012). ToM involves modeling the internal states of others, such as their beliefs, intentions, and differing perspectives, to understand and predict their actions. Grounded in this mental model, an author can then model a reviewer's specific internal state, such as their knowledge background, potential biases, and core concerns, to strategically allocate the limited response space, distinguishing between core critiques that warrant direct rebuttal and minor points that can be tactfully reframed.

In this paper, we propose **RebuttalAgent**, the first model to integrate Theory of Mind into academic rebuttal. RebuttalAgent employs a novel three-stage generation framework we term **_ToM-Strategy-Response (TSR)_**, which decomposes the complex task of rebuttal into a coherent series of reasoning and generation steps. Specifically, the initial Theory-of-Mind (T) stage comprises a hierarchical analysis to discern macro-level reviewer intent while deconstructing the micro-level attributes of each comment. This analysis constructs a multi-dimensional reviewer profile designed to inform both global strategy and local tactics. Subsequently, the Strategy (S) stage utilizes this profile to formulate an actionable plan for the target comment, which aligns the response strategy with both the macro- and micro-level critiques from the reviewer. The process concludes with the Response (R) stage, which achieves evidence-based synthesis by grounding the strategic plan in pre-retrieved evidential chunks to address the inferred reviewer concerns, thereby generating a persuasive response.

To train RebuttalAgent with these complex reasoning capabilities, we construct **RebuttalBench**, a large-scale synthetic dataset of over 70K high-quality samples. This dataset is created via a critique-and-refine pipeline using multiple powerful teacher models, with each sample containing a complete TSR chain. Our training process begins with Supervised Fine-tuning to instill the agent with foundational rebuttal capabilities, and then advances the agent's ToM-based analysis and sophisticated strategic policies via Reinforcement Learning (RL), which is optimized by a novel **_self-reward mechanism_** that enables scalable self-improvement without requiring a separate, external reward model during training. For reliable and efficient automated evaluation, we further develop a specialized evaluator called the Rebuttal-Reward Model (**Rebuttal-RM**). Built upon Qwen3-8B, this model is trained on a diverse, multi-source dataset of over $100K$ samples, which achieves high scoring consistency with human preferences, significantly surpassing the powerful judge GPT-4.1. In summary, our main contributions are as follows:

- We introduce **RebuttalAgent**, the first model to leverage Theory of Mind (ToM) for academic rebuttal, transforming the process from simple linguistic imitation into a strategic reasoning task. Our agent employs a novel ToM-Strategy-Response (TSR) framework that explicitly models the reviewer's perspective, identifies underlying concerns, and formulates evidence-based responses through adaptive strategic reasoning. By bridging the gap between perception and persuasion, RebuttalAgent enables academic communication more clear, constructive.

- We construct **RebuttalBench**, a large-scale dataset of over 70K high-quality samples created via our critique-and-refine pipeline, with each sample containing a ToM-strategy-response chain. Building on the foundational ToM-based reasoning and rebuttal capabilities through SFT, we further optimize the analysis and strategic policies of the agent using RL with our **Self-reward** mechanism, which allowing the agent to continuously improve its persuasive depth and strategic alignment without the need for an external, expert-annotated reward model.

- We develop **Rebuttal-RM**, a specialized evaluator trained on over 100K multi-source samples, achieving a high degree of alignment with human expert preferences, significantly surpassing the consistency of powerful proprietary models like GPT-4.1. Extensive experiments show RebuttalAgent outperforms the base model by an average of 18.3%, and demonstrates performance comparable to advanced proprietary models across both automated and human evaluations.

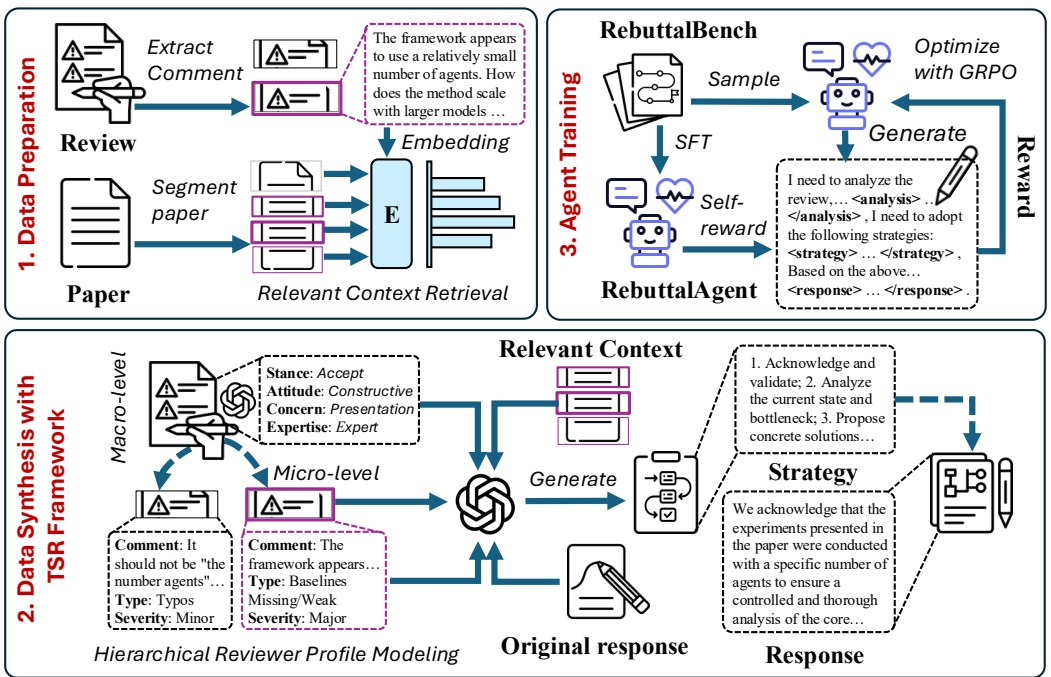

Figure 1: **Overview of our RebuttalAgent framework**. First, we extract each comment from raw reviews and retrieves their relevant context from the paper. Next, based on our TSR framework, we collect a tailored strategy and response for each comment, grounded in Theory of Mind. Finally, our RebuttalAgent is trained via Supervised Fine-Tuning, followed by Reinforcement Learning with a self-reward mechanism, enabling both scalability and self-improvement.

## 2 TASK FORMULATION

In this section, we define the task of academic rebuttal. The core objective is to generate a convincing response to the target comment. Formally, the input of this task consists of:

- **Manuscript** ($M$): The original paper, serving as the evidentiary basis for the rebuttal.
- **Review** ($R_i$): One of $m$ reviews in the set $\mathcal{R} = \{R_1, \ldots, R_m\}$, which contains the specific critiques and queries that must be addressed.
- **Target Comment** ($c_{\text{target}}$): An individual unit of feedback within $R_i$ (e.g., a critique, a query, or an identified weakness) that necessitates a direct response.

Given these inputs, a model $\mathcal{G}$ is tasked with generating a response $r_{\text{target}}$, formalized as:

$$r_{\text{target}} = \mathcal{G}(M, R_i, c_{\text{target}}) \tag{1}$$

The response must be **Convincing**, which goes beyond mere politeness to thoughtfully address the reviewer's concerns and strengthen the paper's position. In addition, it must be deeply **Context-Aware**, demonstrating a nuanced understanding of not only the explicit criticism but also the reviewer's potential underlying assumptions or even misunderstandings. Furthermore, the response must be **Evidence-Grounded**, with every claim and counter-argument verifiably substantiated by the manuscript $M$. Crucially, achieving success lies in the delicate balance of these competing objectives.

## 3 DATA PREPARATION

### 3.1 COMMENT EXTRACTION

Raw reviews often contain a mix of substantive critiques and irrelevant content like greetings or summary restatements. Feeding this unfiltered text directly into a model adds noise and redundancy,

which can reduce the accuracy of the generated rebuttal. Furthermore, due to diverse reviewer writing styles and varying conference formats, comments are typically presented in an unstructured manner. Therefore, to address these challenges and align with our task formulation of addressing a single target comment ($c_{\text{target}}$) at a time, we first process the raw review. Drawing on the powerful information extraction capabilities of LLMs (Zhu et al., 2023; Dagdelen et al., 2024; Schilling-Wilhelmi et al., 2025), we leverage an *LLM-as-Extractor* and design a specific prompt that instructs the LLM to identify and separate each distinct point of criticism from the raw review text to segment a review into discrete, actionable comments. Specifically, the extractor is tasked with decomposing the raw review into a list of original, unedited, critical statements (e.g., "The current analysis lacks a crucial ablation study for component X...making it difficult to ascertain the true contribution."). To validate the reliability of this extractor, we conduct a manual verification on 100 randomly sampled reviews, which achieves a 98% accuracy in comment extraction. The detailed prompt is shown in Appendix E.

## 3.2 CONTEXT RETRIEVAL

A single reviewer comment typically targets a specific aspect of the manuscript, such as a formula or baseline comparison. However, using the full, information-dense manuscript as context is infeasible and sub-optimal, as it can overwhelm the model and dilute focus. Therefore, we implement a three-stage context retrieval module to isolate the most relevant content for each comment. As shown in the top-left corner of Figure 1, the retrieval pipeline begins by segmenting the manuscript ($M$) into discrete text chunks, typically corresponding to paragraphs. Then we employ a pre-trained embedding model[2] to encode both the target comment ($c_{\text{target}}$) and each text chunk into high-dimensional vector representations. Relevance is then quantified by computing the cosine similarity between the comment vector and all chunk vectors. Finally, the top-$k$ chunks with the highest similarity scores are retrieved to serve as the context. The effectiveness analysis of retrieval module is provided in Appendix B.

## 4 TOM-STRATEGY-RESPONSE FRAMEWORK

Theory of Mind (ToM) is a core concept in cognitive science (Leslie et al., 2004; Goldman et al., 2012), referring to the ability to understand and reason about the differing beliefs, intentions, desires, and perspectives of others. Applying this concept to artificial intelligence has led to Machine Theory of Mind (MToM) (Rabinowitz et al., 2018), which is an AI system's capacity to infer and model the mental states of human or AI teammates to support collaboration. Large language models such as GPT-4 have demonstrated stronger ToM-like reasoning capabilities. In our work, we extend MToM to the specific domain of academic rebuttal. Given the game-theoretic and information-asymmetric nature of the rebuttal process, modeling the reviewer's beliefs, knowledge background, and core concerns is particularly critical. Therefore, our proposed RebuttalAgent framework explicitly implements ToM through a Theory-of-Mind-Strategy-Response (TSR) framework, which first constructs a hierarchical reviewer profile to guide the subsequent formulation of strategy and response. Figure 1 (bottom) depicts how our RebuttalAgent framework decomposes the task of rebuttal into a multi-stage reasoning process: (1) inferring the reviewer's perspective with ToM, (2) formulating a tailored strategy, and (3) synthesizing a convincing, evidence-grounded response.

### 4.1 HIERARCHICAL REVIEWER PROFILE MODELING

To capture the underlying intent and stance of reviewers, we propose a hierarchical analysis structure. This structure consists of two levels: a Macro-level analysis to infer the overall intent, which guides the global strategy, and a Micro-level analysis to deconstruct comments for crafting targeted responses.

**Macro-level: Inferring Overall Reviewer Intent.** This analysis employs principles from Theory of Mind to construct a holistic mental model of the reviewer, going beyond the literal text to infer the underlying intent, attitude, and core concerns that subsequently guide the rebuttal's global strategy and tone. We instruct an LLM to interpret the review across four dimensions: Overall Stance, Overall Attitude, Dominant Concern, and Reviewer Expertise, as detailed in Table 4, generating a structured *macro-profile* composed of descriptive categorical labels.

---

[2]https://huggingface.co/Qwen/Qwen3-Embedding-0.6B

**Micro-level: Deconstructing Specific Comments.** This analysis shifts to target comment. We employ an LLM to classify the primary concern of each comment across four key dimensions: Significance, Methodology, Experimental Rigor, and Presentation, as detailed in Table 4. This classification generates a *micro-profile* for each comment. This fine-grained profile enables the formulation of tactical responses that are both precisely targeted and aligned with the global strategy.

## 4.2 ToM-Driven Strategy Generation

The generation of an explicit strategy serves as a crucial intermediate reasoning step, bridging the gap between understanding the reviewer (the profile) and formulating a response. This step translates the static diagnostic profile into a dynamic, actionable plan. To achieve this, the strategy generation process is conditioned on the complete reviewer profile and the target comment itself. We prompt an LLM to synthesize these inputs and output a concise, high-level strategy. The primary benefit of this explicit decomposition is that it compels the LLM to first decide how to respond before writing what to respond. This ensures the final text is not merely reactive to a comment's surface-level query but is strategically aligned with the reviewer's underlying intent, attitude, and primary concerns.

## 4.3 Evidence-Based Strategic Response Generation

The final stage of our TSR framework generates the definitive response ($r_{\text{target}}$) through an advanced guided synthesis process, conditioned on a rich set of strategic and contextual inputs. This intricate process is informed by two distinct yet complementary primary types of input:

- **Strategic Inputs**: The ToM-based reviewer profile ($\mathcal{P}$) and the tailored rebuttal strategy ($S$), which shape how the response engages with the reviewer's likely perspective, guiding its tone and argumentative flow.

- **Contextual Inputs**: The retrieved relevant chunks ($C_E$) and the original response ($r_{\text{orig}}$).

Here, $r_{\text{orig}}$ serves a crucial dual purpose. First, it acts as a high-fidelity source of context, analogous to the retrieved chunks ($C_E$). Second, it provides a high-quality reference for phrasing and structure, which the model uses as a blueprint to refine upon and build the final output.(Notably, $r_{\text{orig}}$ is used only during the data-synthesis phase, not during the final model's inference phase.) Our model, $\mathcal{G}$, generates the response by weaving together these components, ensuring the final text is strategically aligned, factually grounded, and coherently structured. Formally, it is:

$$r_{\text{target}} = \mathcal{G}(\mathcal{R}_i, c_{\text{target}}, \mathcal{P}, S, \bigoplus_{p_j \in C_E} p_j, r_{\text{orig}}) \tag{2}$$

where $\bigoplus$ denotes the concatenation of the text from all relevant chunks in the set $C_E$.

## 5 Agent Training for Strategic Persuasion

### 5.1 RebuttalBench

**(1) Data Source:** Our training data is derived from the $\text{Re}^2$-rebuttal dataset (Zhang et al., 2025), a comprehensive corpus containing initial scientific papers, their corresponding peer reviews, and the authentic author responses. **(2) Data Processing:** The raw data undergoes a multi-stage processing pipeline. First, we utilize GPT-4.1 to parse all the reviews into over 200K distinct comment-response pairs. Following this, each review and comment is annotated with the hierarchical profiles (macro- and micro-level) as defined in Section 4.1. Notably, we explicitly exclude comments that require conducting new, unprovided experiments (e.g., "Compare your method with baseline X"), as we focus the agent's abilities on linguistic persuasion and strategic argumentation, and prevent the model from fabricating or hallucinating experimental data. To ensure a diverse and balanced training set, we then curate a final subset of 70K comments for the next stage, consisting of 60K instances filtered by category and 10K selected randomly. **(3) Data Synthesis:** For each selected comment and its associated authentic response, our **ToM-Strategy-Response (TSR)** framework generates the corresponding *reviewer analysis*, *rebuttal strategy*, and a new, synthetic *response*. To mitigate model-specific biases and enrich stylistic variety, a mixture of powerful teacher models (e.g., GPT-4.1,

Claude 3.5) is used to generate data. To provide the agent with a holistic learning objective, the generated analysis, strategy, and response are structured into a final target sequence. This sequence is a concatenation of the three components, each explicitly demarcated by `<Analysis>`, `<Strategy>`, and `<Response>` tags. Figure D provides a complete example in our RebuttalBench.

## 5.2 Instruction Tuning with ToM-Driven Reasoning

We perform supervised fine-tuning on Qwen3-8B using our RebuttalBench. The objective of this stage is to enable the model to learn the structured reasoning process inherent to the ToM-Strategy-Response framework and to develop its core rebuttal competencies. The diversity of the training data, sourced from varied reviews and synthesized by multiple powerful LLMs, is designed to enhance our agent's robustness and generalization capabilities across different reviewing styles.

## 5.3 Reinforcement Learning with Self-Reward

The former stage equips the agent with the fundamental TSR reasoning. We employ RL to further optimize the agent's outputs to be strategically superior and more convincing.

**Self-Reward Mechanism.** To achieve scalable and self-improving agent capabilities without relying on an externally trained reward model, we introduce a self-reward mechanism. This approach leverages the intrinsic instruction-following and reasoning abilities of the SFT-tuned model $\mathcal{G}_{\text{SFT}}$ to evaluate its own generated outputs autonomously. Specifically, for each candidate output $o$, we assess the response along four critical dimensions. The overall reward is:

$$R(o) = w_1 R_{\text{format}}(o) + w_2 R_{\text{think}}(o) + w_3 R_{\text{resp}}(o) + w_4 R_{\text{div}}(o) \qquad (3)$$

We design multiple reward signals that encourage agent to reason explicitly about various quality dimensions rather than simply restating its prior output. Here, each component is defined as follows: (1) **Format Adherence ($R_{\text{format}}$):** We programmatically check if the output $o$ correctly contains the `<Analysis>`, `<Strategy>`, and `<Response>` structures. This is a binary reward. (2) **Reasoning Quality ($R_{\text{think}}$):** The score is generated by $\mathcal{G}_{\text{SFT}}$ itself. We prompt it to evaluate the quality of the content within the `<Analysis>` and `<Strategy>` blocks, based on criteria such as profiling accuracy and strategic soundness. (3) **Response Quality ($R_{\text{resp}}$):** This score is also generated by $\mathcal{G}_{\text{SFT}}$. We prompt it to evaluate the final `<Response>` content based on persuasiveness, clarity, and the correct use of evidence. (4) **Response Diversity ($R_{\text{div}}$):** To discourage generic and homogeneous outputs and as a mechanism to enhance robustness against reward hacking, we prompt $\mathcal{G}_{\text{SFT}}$ to evaluate a generated `<Response>` content by comparing it against a set of our pre-defined, modular negative samples (i.e., examples of undesirable, templated responses). A higher score is awarded to responses that are semantically distinct from these negative examples, encouraging more varied and human-like replies. The weights $w$ are hyperparameters that balance the contribution of each component. The details of training are provided in Appendix L. We discuss the robustness of our reward signals against reward hacking, particularly focusing on the $R_{\text{div}}$, in Appendix M.

**Optimization Algorithm.** Then, we use the defined rewards to optimize our policy with the Group Reward Policy Optimization (GRPO) algorithm (Guo et al., 2025). For each input question $q$, the model generates a group of $G$ candidates $\{o_1, o_2, \ldots, o_G\}$. The policy $\pi_\theta$ is then updated by optimizing the following clipped surrogate objective:

$$J_{\text{GRPO}}(\theta) = \mathbb{E}\left[\frac{1}{G}\sum_{i=1}^{G} \min\left(\frac{\pi_\theta(o_i|q)}{\pi_{\theta_{\text{old}}}(o_i|q)}A_i, \text{clip}\left(\frac{\pi_\theta(o_i|q)}{\pi_{\theta_{\text{old}}}(o_i|q)}, 1-\epsilon, 1+\epsilon\right)A_i\right) - \beta D_{\text{KL}}(\pi_\theta \| \pi_{\text{ref}})\right] \quad (4)$$

where $\pi_{\theta_{\text{old}}}$ is the policy before the update, $\pi_{\text{ref}}$ is a frozen reference policy for regularization, and $A_i$ is the advantage computed for candidate $o_i$ based on the group's relative rewards.

## 6 Rebuttal-RM as Judge

To conduct both reliable and efficient evaluation, we develop **Rebuttal-RM**, a scoring model specifically trained to automatically assess responses based on the provided target comment and relevant contextual information, with the goal of aligning with human preferences.

Table 1: The consistency scores between various models and the human ratings. We evaluate the models using six standard statistical metrics. Due to space constraints, we present results for only a subset of these metrics in the main paper. More details are provided in Appendix C.1 and Table 11.

| Scoring Model | Attitude | | | Clarity | | | Persuasiveness | | | Constructiveness | | | Avg |
|---|---|---|---|---|---|---|---|---|---|---|---|---|---|
| | $r$ | $\beta$ | $f$ | $r$ | $\beta$ | $f$ | $r$ | $\beta$ | $f$ | $r$ | $\beta$ | $f$ | |
| Qwen3-8B | 0.718 | 0.672 | 0.620 | 0.609 | 0.568 | 0.710 | 0.622 | 0.577 | 0.690 | 0.718 | 0.745 | 0.720 | 0.664 |
| Llama-3.1-8B | 0.297 | 0.347 | 0.540 | 0.158 | 0.047 | 0.380 | 0.272 | 0.245 | 0.560 | 0.424 | 0.457 | 0.460 | 0.349 |
| GLM-4-9B | 0.420 | 0.475 | 0.460 | 0.467 | 0.436 | 0.730 | 0.369 | 0.361 | 0.700 | 0.561 | 0.519 | 0.570 | 0.506 |
| GPT-4.1 | 0.743 | 0.712 | 0.800 | 0.739 | 0.671 | 0.750 | 0.779 | 0.763 | 0.740 | 0.804 | 0.756 | 0.680 | 0.745 |
| DeepSeek-r1 | 0.646 | 0.633 | 0.790 | 0.708 | 0.615 | 0.760 | 0.710 | 0.664 | 0.720 | 0.742 | 0.701 | 0.620 | 0.705 |
| DeepSeek-v3 | 0.699 | 0.733 | 0.710 | 0.687 | 0.578 | 0.740 | 0.697 | 0.652 | 0.770 | 0.771 | 0.719 | 0.750 | 0.692 |
| Gemini-2.5 | 0.620 | 0.509 | 0.750 | 0.605 | 0.593 | 0.540 | 0.627 | 0.607 | 0.520 | 0.711 | 0.705 | 0.610 | 0.616 |
| Claude-3.5 | 0.569 | 0.635 | 0.720 | 0.704 | 0.670 | 0.680 | 0.706 | 0.686 | 0.670 | 0.753 | 0.738 | 0.630 | 0.680 |
| Rebuttal-RM | **0.839** | **0.828** | **0.910** | **0.753** | **0.677** | **0.790** | **0.821** | **0.801** | **0.820** | **0.839** | **0.835** | **0.810** | **0.812** |

**Training Data Construction.** The reward model $\mathcal{G}_{\text{RM}}$ takes the retrieved relevant chunks ($C_E$), the current review $\mathcal{R}_i$, the target comment $c_{\text{target}}$, and a candidate response $r_{\text{target}}$ as input. It outputs a set of multi-dimensional scores, $\mathbf{s}$, and an explanation, $e$. This process is formalized as:

$$(\mathbf{s}, e) = \mathcal{G}_{\text{RM}}\Big( \bigoplus_{p_j \in C_E} p_j, \mathcal{R}_i, c_{\text{target}}, r_{\text{response}} \Big) \tag{5}$$

We construct a dataset of over 102K instances from three sources: (1) 12,000 original author responses as a realistic human baseline, (2) high-quality GPT-4.1-refined responses representing top standards, and (3) diverse model-generated replies (e.g., Qwen2.5-3B, Claude 3.5) for style coverage. To acquire the ground-truth labels $(\mathbf{s}, e)$ for these inputs, we employ a hybrid annotation strategy. For the original author responses, instances where the reviewer subsequently raises their score are considered high-quality, and these are then manually scored by our team. For the responses generated by various models, we utilize Gemini 2.5 Pro to automatically generate the corresponding scores and explanations. Detailed statistics are provided in Table 10.

**Rebuttal-RM Training** We use 90% of above labeled data for training and 10% for testing. We select Qwen3-8B as the base model and fine-tune it on our constructed training dataset to create the final Rebuttal-RM. The details of training and evaluation is in Appendix K.

## 7 EXPERIMENT

### 7.1 EVALUATION OF REBUTTAL-RM

Following recent work (Wu et al., 2025), we employ a set of six statistical metrics to validate the effectiveness of Rebuttal-RM. We use four standard statistical measures to assess the overall correlation: *Mean Absolute Error (e), Pearson (r), Spearman (β), and Kendall (τ)*. Additionally, to mitigate potential annotator biases and assess classification accuracy, we introduce two metrics based on score ranges: *Coarse-grained Accuracy (c)*: Scores are mapped to four quality tiers: Unconvincing (scores 1-3), Acceptable (scores 4-6), Good (scores 7-8), and Excellent (scores 9-10). *Fine-grained Accuracy (f)*: For a stricter assessment, scores are categorized into seven more granular ranges derived from our rubric, such as grouping scores of 1 and 2, 3 and 4, and so on.

**Rebuttal-RM Aligns Better with Human Evaluators.** Table 1 shows that Rebuttal-RM outperforms all baselines in alignment with human judgments, achieving the highest average score (0.812) and leading in all individual metrics. Notably, it surpasses GPT-4.1 and DeepSeek-r1 by 9.0% and 15.2%, respectively. Full results are provided in Appendix Table 11.

### 7.2 BENCHMARKING REBUTTALAGENT

**Baselines.** We evaluate our RebuttalAgent against two categories of baselines: foundation models and agent-based methods. (1) The **Foundation Models** include o3, GPT-4.1 (Hurst et al., 2024), Deepseek-R1 (Guo et al., 2025), Deepseek-V3 (Liu et al., 2024a), Gemini-2.5 (Comanici et al., 2025), GLM-4-9B (GLM et al., 2024), Llama-3.1-8B-Instruct (Grattafiori et al., 2024), and Qwen3-8B

Table 2: Performance comparison of RebuttalAgent with baseline models and ablation study results on R2-test. Due to space constraints, we only present $C$, $P$, and $Co$. For complete results, please refer to Table 8. For the ablations, w/o indicates the removal of a specific reward component (e.g., w/o $R_{\text{reasoning}}$), while w/ Distinct Weights indicates the use of distinct reward weights. The delta values ($\Delta$) reported in the table are computed with respect to the base model.

| Category | Rigor | | | Soundness | | | Significance | | | Presentation | | | Avg |
|---|---|---|---|---|---|---|---|---|---|---|---|---|---|
| Metric | C | P | Co | C | P | Co | C | P | Co | C | P | Co | |
| o3 | 9.00 | 8.99 | 9.55 | 8.84 | 8.78 | **9.45** | 8.58 | 8.43 | 9.22 | 9.34 | 9.12 | 9.50 | 9.21 |
| GPT-4.1 | 8.34 | 7.86 | 8.80 | 8.27 | 7.79 | 8.62 | 8.05 | 7.28 | 8.20 | 8.91 | 8.57 | 9.42 | 8.50 |
| DeepSeek-R1 | 8.47 | 7.90 | 8.90 | 8.46 | 8.03 | 8.75 | 8.29 | 7.71 | 8.60 | 9.03 | 8.70 | **9.54** | 8.64 |
| Deepseek-V3 | 8.43 | 7.67 | 8.83 | 8.42 | 7.71 | 8.72 | 8.18 | 7.35 | 8.59 | 8.94 | 8.45 | 9.41 | 8.51 |
| Gemini-2.5 | 7.89 | 6.91 | 6.63 | 8.06 | 7.41 | 7.26 | 7.87 | 6.99 | 6.89 | 8.56 | 8.11 | 8.83 | 7.75 |
| GLM-4-9B | 8.08 | 7.46 | 8.69 | 7.97 | 7.24 | 8.26 | 7.84 | 6.90 | 8.11 | 8.52 | 8.02 | 8.99 | 8.13 |
| Llama-3.1-8B | 7.77 | 6.69 | 7.32 | 7.71 | 6.76 | 7.02 | 7.54 | 6.30 | 6.49 | 8.12 | 7.42 | 8.25 | 7.44 |
| Qwen3-4B | 7.84 | 7.05 | 7.42 | 7.77 | 6.98 | 6.99 | 7.72 | 6.69 | 6.83 | 8.48 | 8.02 | 8.66 | 7.69 |
| Qwen3-8B | 7.96 | 7.33 | 8.18 | 7.84 | 7.11 | 7.76 | 7.68 | 6.73 | 7.39 | 8.51 | 8.08 | 8.87 | 7.96 |
| Self-Refined | 8.55 | 8.08 | 9.04 | 8.47 | 8.04 | 8.88 | 8.19 | 7.56 | 8.52 | 9.08 | 8.75 | 9.59 | 8.72 |
| Strategy-Prompt | 8.26 | 7.41 | 8.32 | 8.33 | 7.77 | 8.51 | 8.13 | 7.41 | 7.95 | 8.85 | 8.44 | 9.46 | 8.37 |
| TSR$_{o3}$ | 8.89 | **9.10** | 9.68 | 8.95 | 8.91 | 9.28 | 8.69 | **8.56** | 9.45 | 9.18 | **9.35** | 9.45 | 9.34 |
| TSR$_{\text{GPT4.1}}$ | 8.47 | 7.63 | 8.53 | 8.12 | 7.74 | 8.85 | 7.90 | 7.51 | 8.45 | 9.07 | 8.42 | 9.16 | 8.76 |
| RebuttalFT | 6.91 | 6.07 | 6.80 | 6.58 | 5.72 | 6.24 | 6.52 | 5.50 | 5.94 | 6.55 | 5.79 | 6.63 | 6.35 |
| RebuttalAgent | **9.23** | 8.91 | **9.59** | **9.18** | **8.95** | 9.37 | **9.09** | 8.54 | **9.65** | **9.43** | 9.20 | 9.50 | **9.42** |
| $\Delta$ ($\uparrow$) | 16.1% | 21.6% | 22.1% | 17.0% | 25.9% | 28.4% | 18.3% | 26.9% | 34.6% | 10.8% | 13.8% | 12.6% | 18.3% |
| *Data Ablation* | | | | | | | | | | | | | |
| w/o ToM | 8.91 | 8.21 | 9.29 | 8.88 | 8.30 | 9.28 | 8.70 | 7.87 | 9.38 | 9.22 | 8.86 | 9.58 | 9.04 |
| w/o Strategy | 9.01 | 8.89 | 9.93 | 9.00 | 8.85 | 9.30 | 8.88 | 8.49 | 9.82 | 9.27 | 9.06 | 9.33 | 9.31 |
| w/o Thinking | 9.06 | 9.00 | 9.18 | 9.02 | 8.92 | 9.13 | 8.96 | 8.60 | 9.20 | 9.35 | 9.16 | 9.55 | 9.37 |
| *Training Ablation* | | | | | | | | | | | | | |
| w DPO | 8.47 | 8.13 | 9.36 | 8.32 | 7.92 | 9.00 | 8.11 | 7.57 | 8.82 | 8.94 | 8.55 | 9.46 | 8.68 |
| SFT-only | 8.20 | 7.60 | 8.42 | 8.17 | 7.60 | 8.28 | 8.02 | 7.31 | 7.84 | 8.76 | 8.34 | 9.16 | 8.27 |
| RL-only | 8.63 | 8.27 | 9.42 | 8.47 | 8.07 | 9.01 | 8.21 | 7.56 | 8.34 | 9.05 | 8.71 | 9.61 | 8.79 |
| w/o R$_{\text{Analysis}}$ | 9.25 | 9.23 | 9.79 | 9.20 | 9.18 | 9.39 | 9.00 | 8.87 | 9.27 | 9.59 | 9.41 | 9.45 | 9.23 |
| w/o R$_{\text{Response}}$ | 8.51 | 7.90 | 9.02 | 8.41 | 7.91 | 8.63 | 8.17 | 7.51 | 8.25 | 9.05 | 8.68 | 9.61 | 8.63 |
| w/o R$_{\text{Format}}$ | 9.06 | 8.91 | 9.22 | 9.04 | 8.74 | 9.30 | 8.88 | 8.29 | 9.67 | 9.37 | 9.14 | 9.35 | 9.32 |
| w R$_{\text{Dist. weights}}$ | 9.08 | 8.54 | 9.53 | 9.04 | 8.63 | 9.23 | 9.05 | 8.32 | 9.85 | 9.34 | 9.08 | 9.38 | 9.27 |
| w RebuttalRM-reward | 9.39 | 9.35 | 9.51 | 9.40 | 9.32 | 9.29 | 9.53 | 8.95 | 9.70 | 9.61 | 9.45 | 9.89 | 9.45 |
| w GPT4.1-reward | 9.33 | 9.24 | 8.85 | 9.32 | 9.16 | 9.82 | 9.35 | 9.07 | 9.30 | 9.24 | 9.38 | 9.18 | 9.35 |
| w Llama-based | 9.23 | 9.10 | 9.16 | 9.29 | 9.11 | 9.24 | 9.16 | 8.67 | 9.05 | 9.57 | 9.35 | 9.39 | 9.20 |
| w Qwen3-4B-based | 8.79 | 8.54 | 9.73 | 8.60 | 8.24 | 9.44 | 8.32 | 7.84 | 9.17 | 9.12 | 8.76 | 9.72 | 8.98 |

(Yang et al., 2025a). (2) The **Agent-based Methods** comprise three distinct approaches, with the first two leveraging GPT-4.1 as the backbone model: *Self-Refined*, which generates an initial response and then iteratively refines it via self-reflection; *Strategy-Prompt*, which mimics our methodology by first generating a strategic plan based on an analysis of reviewer comments before writing the final rebuttal; and *RebuttalFT*, a Qwen3-8B model directly supervised fine-tuned on the $R^2$-rebuttal dataset, which contains real-world, human-written rebuttals.

**Metrics.** Our primary metric is a holistic quality score on a scale of 0-10, where a higher score indicates a superior response, ranging from *Wholly Ineffective* (0) to *Outstanding* (9-10). This holistic score is supported by a breakdown into four key dimensions, each also rated on a 0-10 scale: **Clarity (C)** (logical flow and organization), **Persuasiveness (P)** (argument strength and evidence), and **Constructiveness (Co)** (commitment to improvement and actionable revisions), **Attitude (A)** (tone and professionalism). These criteria form the rubric for our Rebuttal-RM automated evaluation, enabling our Rebuttal-RM to provide not only an overall quality score but also interpretable diagnostics.

**Datasets.** (1) In-domain test set, R2-test, contains 6,000 comments randomly sampled from the Re$^2$ dataset (Zhang et al., 2025), with no training data overlap. Sourced from 24 conferences and 21 workshops on OpenReview (2017–2023), it offers broad topic and style diversity, enabling comprehensive evaluation of familiar academic discourse. (2) For out-of-domain evaluation, we introduce Rebuttal-test. We manually collect over one thousand recent ICLR and NeurIPS reviews (post-2023) from OpenReview, ensuring no data overlap with our training set or R2-test. These reviews are then processed using the comment extraction and context retrieval pipeline, resulting in a final set of 2K comments designed to assess generalization capability.

## 7.3 EXPERIMENTAL RESULTS

**RebuttalAgent Significantly Outperforms Baselines.** As shown in Table 2, our RebuttalAgent achieves the highest overall average score of **9.42**, substantially outperforming all baselines including

GPT-4.1 and o3. It excels across key rebuttal dimensions, attaining top Clarity (9.43) and strong Persuasiveness (9.20) scores. Compared to the Qwen3-8B baseline, the agent yields an average improvement of **18.3%**, with the most significant gains in Persuasiveness and Constructiveness (up to 34.6%). Full results on R2-test are provided in Table 8, while the out-of-domain evaluation (i.e., results on our constructed Rebuttal-test) is presented in Table 9.

**Ablation Study.** Our ablation study confirms the necessity of all the model's design components. Performance significantly drops when removing any key component, such as ToM, Strategy, Thinking, or when omitting core training stages such as SFT and RL. Among all reward signals, the one for final response quality proved to be the most impactful. These results show that our model's success is rooted in the synergy between its specialized data, complete training process, and reward mechanism. Applying our framework to Llama-3.1-8B and Qwen3-4B yields significant gains, raising scores from 7.44 to 9.20 and 7.69 to 8.98, respectively. These results demonstrate that the effectiveness of our TSR framework and self-reward mechanism is not tied to a specific backbone, it serves as a model-agnostic strategy that generalizes well to other models, including smaller ones.

**Effectiveness of ToM-Driven Reasoning.** To evaluate the effectiveness of our ToM-Driven reasoning, we use the ToM analysis and Strategy generated by our model as contextual input for two external base models: Qwen3-8B and Llama3.1-8B. Results in Figure 2 show consistent performance gains across all data categories for both models, confirming our approach's robustness and transferability. The most significant improvements

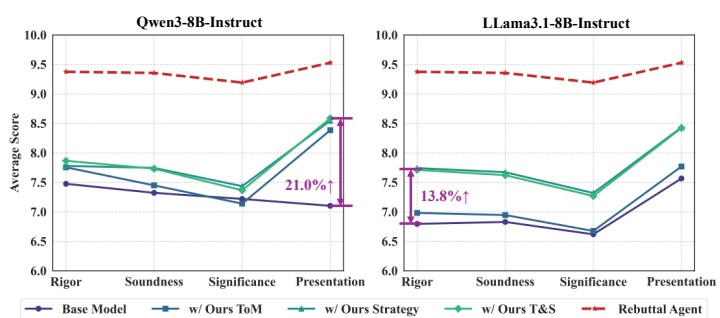

Figure 2: Performance of base models when augmented with the ToM analysis and Strategy generated by our model.

arise when using our Strategy or the full ToM and Strategy (T&S) as context. Notably, Qwen3-8B achieves a **21.0%** gain in **Presentation** when guided by the full T&S. Nevertheless, the complete RebuttalAgent maintains a clear performance edge, suggesting that optimal efficacy is achieved within its fully integrated architecture. Results are available in Table 7.

## 7.4 HUMAN EVALUATION

We perform a human evaluation as our gold-standard assessment, with detailed results presented in Table 3. The evaluation utilizes a set of 100 randomly sampled comments, balanced between in-domain and out-of-domain instances. Each response is evaluated blindly by three annotators with at least three years of research experience in AI/ML and prior reviewing experience in top-tier conferences on a 10-point scale across four distinct metrics. The reliability of this process is underscored by a strong inter-annotator agreement (Cohen's $\kappa = 0.79$).

**Result.** As presented in Table 3, the human evaluation results decisively confirm the clear superiority of RebuttalAgent. Our model achieves the highest average score of 9.57, establishing a significant lead over all the strongest baselines, o3 and GPT-4.1. This advantage is comprehensive, as RebuttalAgent outperforms all other models across all four evaluation dimensions.

Table 3: Human evaluation results based on four evaluation metrics: Attitude, Clarity, Persuasiveness, and Constructiveness.

| Metric | Attitude | Clarity | Persuasiveness | Constructiveness | Avg |
|---|---|---|---|---|---|
| o3 | 9.30 | 9.28 | 9.04 | 9.42 | 9.26 |
| GPT-4.1 | 9.32 | 8.80 | 8.70 | 9.14 | 8.99 |
| DeepSeek-R1 | 9.24 | 9.08 | 8.86 | 9.16 | 9.08 |
| Qwen3-8B | 8.88 | 8.60 | 8.12 | 8.40 | 8.50 |
| w GPT4.1-reward | 9.92 | 9.62 | 9.28 | 9.54 | 9.59 |
| w RebuttalRM-reward | 9.16 | 8.90 | 8.84 | 9.07 | 8.96 |
| RebuttalFT | 7.38 | 6.80 | 6.30 | 6.50 | 6.75 |
| RebuttalAgent | **9.86** | **9.38** | **9.34** | **9.68** | **9.57** |

Our RebuttalAgent demonstrates the largest relative gain in Persuasiveness, achieving a score of 9.34 which represents a **7.36%** improvement over the GPT-4.1 baseline. This finding, combined with high scores in other metrics, confirms that RebuttalAgent by far is the most effective and balanced model.

## 8 RELATED WORK

**Machine Theory of Mind.** Machine Theory of Mind (ToM) refers to an AI system's capacity to infer and model the mental states of human or AI teammates to support collaboration (Rabinowitz et al., 2018; Goldman et al., 2012; Wellman, 2002; Huang et al., 2025; Yang et al., 2025c; Leslie et al., 2004). Instruction-tuned models such as GPT-4 have demonstrated stronger ToM-like reasoning compared to earlier versions (Kosinski, 2023; 2024), sometimes matching or exceeding human performance in tasks involving sarcasm and social inference. Various methods have been proposed to explicitly model ToM. For example, SymbolicToM builds symbolic belief graphs to track character beliefs for answer generation (Sclar et al., 2023). SimToM employs perspective-taking and context filtering in a two-stage process (Wilf et al., 2023), while ToM-LM translates questions into symbolic forms for model checking (Tang & Belle, 2024). ToMAP integrates opponent modeling and reinforcement learning to generate more persuasive arguments (Han et al., 2025). In this work, we extend machine ToM to the academic rebuttal and propose RebuttalAgent, which operationalizes ToM through a hierarchical analysis of reviewer intent and a multi-stage reasoning pipeline.

**LLM Debate.** The use of multi-agent debate and interaction among Large Language Models (LLMs) has emerged as a promising approach to enhance capabilities in complex reasoning (He et al., 2023; Xu et al., 2024b; He et al., 2024a; Ju et al., 2026; Yan et al., 2025; Shen et al., 2025; Qin et al., 2025; Xu et al., 2024a; Yang et al., 2025b; He et al., 2025d; Chen et al., 2025c) and fact-checking by simulating collaborative or adversarial dialogue (Du et al., 2023; He et al., 2025a; Ma et al., 2024; Chen et al., 2024; Liang et al., 2023; Jin et al., 2024a; Breum et al., 2024; He et al., 2025c; Salvi et al., 2025). For instance, ChatEval employs a multi-agent referee team to evaluate open-ended responses (Chan et al., 2023), while AgentsCourt improves answer quality through multi-round debate among model instances (He et al., 2024b). Debatrix provides a structured judging framework to assess debates along multiple dimensions (Liang et al., 2024), and DyLAN dynamically assembles agent teams tailored to different tasks (Liu et al., 2024c). Notably, Salvi et al. (2025) shows that GPT-4 equipped with sociodemographic data can outperform humans in persuasion. In this work, we integrate Theory of Mind to strengthen the strategic reasoning and persuasive capacity of LLMs in academic rebuttals, aiming to foster more transparent and constructive scholarly dialogue.

**LLM for Academic Peer Review.** The emerging paradigm of AI for Research applies Large Language Models (LLMs) to automate and enhance scholarly activities, including automated research (Schmidgall & Moor, 2025; Li et al., 2025; Yamada et al., 2025) and writing assistance (Wang et al., 2025; Chen et al., 2025a). Within the critical domain of peer review, LLMs are leveraged for generating reviews (Zhu et al., 2025; Idahl & Ahmadi, 2025) and for enhancing review quality analysis (Purkayastha et al., 2025). Furthermore, multi-agent systems have been proposed to explore peer review dynamics (Jin et al., 2024b; D'Arcy et al., 2024) and automate research workflows (Schmidgall et al., 2025). Despite the creation of large, multi-turn review datasets (Zhang et al., 2025), there remains limited exploration into the rebuttal stage. Building on these foundations, our work proposes a RebuttalAgent framework that explicitly leverages Theory of Mind to model reviewer intent, enabling more strategic and context-aware responses.

## 9 CONCLUSION

In this paper, we introduce RebuttalAgent, the first framework to ground academic rebuttal in Theory of Mind (ToM), transforming the process into a strategic reasoning task. By integrating evidence-based synthesis with hierarchical reviewer profiling, our model move beyond formulaic templates to facilitate more constructive scholarly dialogue. To support this, we construct RebuttalBench, a large-scale dataset of 70K reasoning chains, and optimize the agent via reinforcement learning with a Self-reward mechanism. Furthermore, our specialized evaluator, Rebuttal-RM, demonstrates superior alignment with human expert preferences compared to proprietary models. Extensive experiments show that RebuttalAgent outperforms base models by 18.3%, achieving performance comparable to advanced models like o3 across both automated and human evaluations.

## ETHICAL CONSIDERATION

We introduce a comprehensive framework agents for the academic rebuttal process. The goal of this work is to improve the clarity and constructive nature of academic dialogue. The resulting tool is intended to serve as a valuable reference and guidance resource for fresh scholars, offering strategic suggestions and practical tips to help them navigate this complex stage more effectively, rather than as a replacement for genuine scholarly engagement. While RebuttalAgent can clarify the organization and articulation of rebuttals, it is important to recognize its limitations. Like other AI systems, RebuttalAgent may inadvertently learn and reinforce biases present in its training data, such as inappropriate and unscholarly persuasion strategies or rebutting evidence. To mitigate misuse, we specifically excluded comments related to experimental results during training, thus preventing the model from fabricating evidence or data. Authors must view the generated output critically to ensure the accuracy, fairness, and rationality of the generated context. Ultimately, our vision is for RebuttalAgent to serve as a powerful AI assistant for researchers in any field, helping to facilitate more effective human-AI collaboration and foster a more open and constructive scientific world.

## REPRODUCIBILITY STATEMENT

This framework comprises three main components: (1) a rebuttal evaluator, **Rebuttal-RM**; (2) a large-scale high-quality dataset, **RebuttalBench**; and (3) a novel academic assistant, **RebuttalAgent**. To ensure the full reproducibility of this framework, we have provided detailed documentation across the paper and its appendices. The generation process for the RebuttalBench dataset, along with the complete training procedures for RebuttalAgent (including all hyperparameters), are provided in Section 5. The details for training Rebuttal-RM are provided in Section 6. Our code and models are publicly available at https://github.com/Zhitao-He/RebuttalAgent.

## ACKNOWLEDGMENTS

This work was supported in part by grants UROP26EG06 and WEB26EG02, Hong Kong University of Science and Technology. We would like to express our sincere gratitude to the anonymous reviewers for their insightful comments and constructive suggestions.

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

## A   LLM USAGE

This paper introduces a comprehensive framework for leveraging Theory of Mind (ToM) for academic rebuttal, resulting in the **RebuttalAgent**, the **RebuttalBench** dataset, and the **Rebuttal-RM** evaluator. In the preparation of this manuscript, we utilized Large Language Models (e.g., Google's Gemini and GPT-4.1) as a general-purpose writing assistant. The scope of the LLM's assistance was limited to language-level polishment. This included a number of specific tasks: detecting and correcting grammatical and syntactical mistakes; giving suggestions on substitute phrasing to improve sentence flow and coherence; enhancing vocabulary for better precision and stylistic consistency; and paraphrasing author-written sentences to improve readability and prevent repetition.

## B   DATA PREPARATION

**Comment Extraction Accuracy**: To assess the accuracy of our comment extraction approach, we randomly sampled 100 raw reviews and manually examined the extracted comments. Each extracted comment was checked to determine whether it accurately captured a distinct and actionable criticism from the original review. Our analysis shows that over 98 percent of the extracted comments were both complete and well-aligned with the reviewers' intended points, while only 2 percent of the comment contained minor segmentation errors or incorporated redundant information. These results demonstrate the robustness of our LLM-as-Extractor framework in handling diverse reviewer writing styles and unstructured review formats.

**Context Retrieval Effectiveness** We conduct a comprehensive evaluation of our context retrieval pipeline by comparing different retrieval and manuscript segmentation strategies. Specifically, we evaluat three comment encoding strategies: (1) directly using the original comment for retrieval, (2) rewriting the comment from the reviewer's perspective before retrieval, and (3) rewriting the comment from the author's perspective. For manuscript segmentation, we compare splitting the text into 80 parts by word count, 60 parts by word count, and segmenting solely by paragraph. Cosine similarity is employed as the primary quantitative metric to assess retrieval effectiveness across all settings. As illustrated in Figure 3, the results show that using the original comment directly as the retrieval query, combined with segmenting the manuscript by paragraph, achieves the highest retrieval effectiveness. This configuration yields superior performance compared to alternative combinations, highlighting the importance of both precise comment formulation and natural document segmentation.

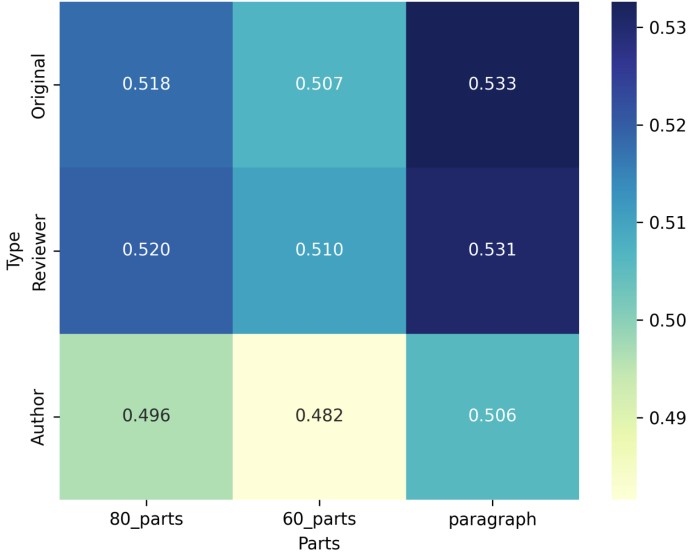

Figure 3: Heatmap for retrieval effectiveness

Table 4: Dimensions of the Hierarchical Reviewer Profile. The complete list of categories, along with a visualization of the data distribution for reviews and comments, is provided in Appendix C.

| Dimension | Description | Example Categories |
|---|---|---|
| *Macro-level* | | |
| Overall Stance | Predicts the reviewer's likely final recommendation on the manuscript. | Reject, Accept |
| Overall Attitude | Assesses the underlying sentiment and tone. | Constructive, Skeptical |
| Dominant Concern | Identifies the primary area of criticism. | Methodology, Experiments |
| Reviewer Expertise | Estimates the reviewer's topic familiarity. | Domain Expert, Generalist |
| *Micro-level* | | |
| Significance | Identifies concerns about impact or novelty. | Incremental, Unclear |
| Methodology | Pinpoints flaws in the technical approach. | Technical Error, Unjustified |
| Experimental Rigor | Addresses issues related to the soundness of the empirical validation. | Baselines Missing, Flawed |
| Presentation | Flags issues related to clarity and structure. | Writing Issues, Poor Org. |

## C  DISTRIBUTION OF REVIEWS AND COMMENTS

### C.1  SETUP AND METRICS OF REBUTTAL-RM

Following recent work (Wu et al., 2025), we employ a set of six statistical metrics. We use four standard statistical measures to assess the overall correlation: *Mean Absolute Error (e), Pearson (r), Spearman ($\beta$), and Kendall ($\tau$)*. Additionally, to mitigate potential annotator biases and assess classification accuracy, we introduce two metrics based on score ranges: *Coarse-grained Accuracy (c)*: Scores are mapped to four quality tiers: Unconvincing (scores 1-3), Acceptable (scores 4-6), Good (scores 7-8), and Excellent (scores 9-10). *Fine-grained Accuracy (f)*: For a stricter assessment, scores are categorized into seven more granular ranges derived from our rubric, such as grouping scores of 1 and 2, 3 and 4, and so on, with single-point ranges for scores of 5 and 6.

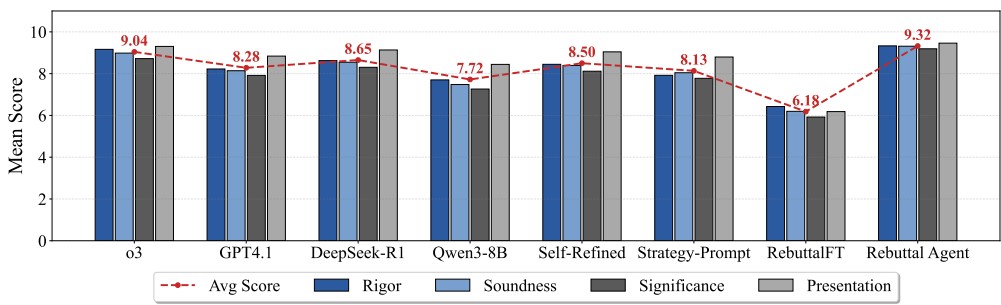

Figure 4: Comparative evaluation of model performance on rebuttal quality.

## D  INSTRUCTION FOR SFT WITH OUTPUT FORMAT EXAMPLE

You are an expert academic assistant specializing in crafting persuasive and respectful rebuttals for peer reviews. Your goal is to formulate a response that addresses the reviewer's concerns directly and constructively, ultimately strengthening the paper's position for acceptance.

You receive the following inputs:
1. Full_Review_Content: The entire review text for the target paper.

2. Target_Comment: A specific excerpt from the review that requires a response.

3. Relevant_Paper_Fragment: A key excerpt from the author's own manuscript. This fragment provides the essential context and technical details that relevant to the Target_Comment.

Your task is to generate a structured rebuttal plan and response by following these steps precisely:

**Step 1: Analysis**

First, conduct your analysis of the overall review and target comment. Present this analysis inside <analysis> and </analysis> tags using the strict JSON format specified below.

**Step 2: Rebuttal Strategy**

Based on your analysis and the information within the Relevant_Paper_Fragment, devise an optimal, step-by-step strategy for the response. Present this strategy as a numbered list inside <strategy> and </strategy> tags. Each step should be a clear action.

Otherwise, omit this section.

**Step 3: Rebuttal Response**

Finally, craft the rebuttal response for the Target_Comment. Write the response inside <response> and </response>, based on your above analysis and strategy.

Here is an example of output format:

```
I need to analysis the review's overall instance and the target
    ↪ comment:
<analysis>{
  "global\_profile": {
    "overall\_stance": "...",
    "overall\_attitude": "...",
    "dominant\_concern": "...",
    "reviewer\_expertise": "..."
  },
  "comment\_analysis":
    {
      "comment\_text": "...",
      "category": "...",
      "sub\_category": "...",
      "severity": "..."
}}
</analysis>.Based on current overall analysis, to address the target
    ↪ comment, I need to adopt the following strategies:
<strategy> 1. ; 2. ; 3. ; XXX</strategy>.
```

Based on the above analysis and strategies, for the target comment:

```
<response>XXX</response>.
```

# E    INSTRUCTION FOR SFT SCORING MODEL WITH OUTPUT FORMAT EXAMPLE

You are a seasoned academic reviewer and response optimization expert. Your task is to evaluate the quality of the response based on the review comments, paper fragments, and the authors' responses. Please strictly follow the requirements below, and output only the score

and score explanation.

Input variables:
1.Full_Review_Content : Complete content of the review comments.
2.Relevant_Paper_Fragment: The paper fragment most relevant to the comment.
3.Target_Comment: Specific segment of the review comments.
4.Original_response: The authors' original response text to the comment.

Your task: Based on the input information, output only a JSON object containing the following two items: Scoring Standard: Score Range: 0 - 10
0: Wholly Ineffective 1-2: Perfunctory 3-4: Unconvincing 5-6: Addresses Some Concerns 7-8: Exceptional 9-10:Outstanding
**Four-dimensional score breakdown, ranging from 0-10, structured as follows:**
Attitude: The tone and professionalism of the response.
Clarity: The logic, structure, and focus of the response.
Persuasiveness: The effectiveness of the argumentation and evidence support.
Constructiveness: The commitment to revisions and specific actions taken.
ScoreExplanation: A brief explanation of each score, specifically citing key points from the response text that reflect the scores and any shortcomings.

**Output requirements:**
Only output the JSON object; do not include any other characters or explanations. The scoring must be reasonable, and the score explanation must clearly reference the original text that reflects the score. All output must be in formal, polite academic English.
Output format example:

```
{ "score": {  "Attitude": <int>,
               "Clarity": <int>,
               "Persuasiveness": <int>,
               "Constructiveness": <int> },
"score\_explanation": <explanation for your given score> }
```

## F  PROMPT FOR REVIEWER STANCE MODELING

You are a world-class AI assistant specializing in the meta-analysis of academic peer reviews. Your task is to act as an experienced and insightful scholar, dissecting a reviewer's comments with extreme precision and objectivity. Your ultimate goal is to perform a comprehensive two-level analysis (Macro and Micro) on the provided review text and output a SINGLE, VALID JSON object that encapsulates your findings. Do not add any explanatory text, comments, or markdown formatting like "'json before or after the JSON output.
**EXECUTION STEPS**:
**Macro-Analysis**:
Read the entire review text holistically. Determine the four macro-level attributes: Overall Stance, Overall Attitude, Dominant Concern Theme, and Reviewer Expertise Proxy.
**Micro-Analysis**:

Extract all distinct reviewer questioning opinions, weaknesses, shortcomings, criticisms, and actionable suggestions for improvement.

**Key Section Focus**:

Search for and extract content specifically from sections likely to contain negative feedback, issues, or suggestions (e.g., Summary of Weaknesses, Weaknesses, Comments Suggestions And Typos, Comments, Critiques, Suggestions, Detailed Feedback, Concerns, Issues, Discussion Points, or other similar sections).

**Extraction Rule**:

Treat each numbered item (e.g., 1., 2.) or bullet point as a single, unified reviewer comment, even if it contains multiple ideas or sub-points. Do not split such items further. For vaguely phrased or ambiguous sentences, distill them into clear, distinct opinions without altering their original intent.

**Strictly Exclude**:

Any positive feedback (e.g., content from Summary of Strengths or similar sections). Any meta-comments about the review process or reviewer confidence (e.g., Confidence, Soundness, Excitement, Overall Assessment, etc.).

**For each extracted reviewer comment**:

Classify it into one main category and its corresponding sub-category (see KEY DEFINITIONS). Assign a severity level. Assign an API model confidence score (see below). Populate the final JSON object strictly according to the definitions and schema provided below.

**KEY DEFINITIONS**:

**Macro-Analysis Definitions:**

**Overall Stance Prediction:**

Accept: Clear intention to accept.

Probably Accept: Leaning towards acceptance, but with some reservations.

Borderline: Reviewer is undecided; the decision could go either way.

Probably Reject: Leaning towards rejection, but might be convinced by a strong rebuttal.

Reject: Clear intention to reject.

Note: Reference any given rating/confidence if present, otherwise infer from reviewer language.

**Overall Attitude Assessment:**

Enthusiastic: Strong positive language, focuses on strengths.

Constructive: Balanced, flaw-pointing with intent to help improve.

Neutral: Report-like, factual, little emotional language.

Skeptical: Questioning, challenging, demanding proof

Dismissive: Strong negative language, pre-judged against the paper.

**Dominant Concern Theme:**

Novelty & Significance

Methodological Soundness

Experimental Rigor

Presentation & Clarity

**Reviewer Expertise Proxy:**

Domain Expert

Generalist

Unfamiliar

**Micro-Analysis Definitions**:

**Categories and Sub-categories:**

**Novelty and Significance:**

Contribution Unclear

Incremental Contribution

Motivation Weak

**Methodological Soundness:**

Technical Error

Unjustified Assumption

Lack of Detail

**Experimental Rigor:**

Baselines Missing/Weak

Insufficient Experiments

Ablation/Analysis Missing

Flawed Evaluation

**Presentation and Clarity:**

Writing Issues/Typos

Poor Organization

Figure/Table Quality

Related Work Incomplete

**Severity:**

Major: Requires substantial work to fix (e.g. new experiments).

Minor: Can be fixed with modest effort (e.g. rewriting a paragraph, fixing a figure).

**API Model Confidence Global_Profile and Micro_Analysis:**

For each global_profile and micro_level comment, output the AI model's own confidence in its classification of category, sub-category, and severity.

Use a score from 1 to 10 where:

10: Extremely confident (review statement is explicit and unambiguous)

5: Moderate confidence (some ambiguity or open to interpretation)

1: Very low confidence (classification is highly uncertain due to vagueness or lack of detail)

**EXTRACTION GUIDELINES (CRUCIAL):**

Only extract criticism, questions, actionable feedback, and suggestions for improvement.

Do NOT extract any positive feedback, praise, or general statements of merit.

Do NOT include meta-comments about the review process or reviewer confidence.

Each numbered or bullet-pointed item should be treated as a single, indivisible comment, even if it contains multiple ideas. For ambiguous sentences, distill them into clear, distinct opinions without altering the original intent.

**Here is a extraction example:**

Summary of Strengths:

1. The authors conduct detailed experiments across several editing tasks and metrics, with comparisons against multiple SOTA baselines. 2. MedEBench provides fine-grained editing categories, quantitative and qualitative ground truths, and a protocol that reflects real clinical scenarios. 3. The paper provides useful observations about the limitations of current models in medical contexts, especially in preserving anatomical structures and semantic consistency.

Summary of Weaknesses:

1. The paper notes that editing bones is more challenging for current models, but does not provide detailed analysis or hypotheses for why this might be the case. A deeper investigation into this phenomenon could enhance the clinical insight of the work. 2. Given the medical setting, there should be discussion on privacy implications, especially concerning synthetic

patient-like data. For instance, could generated images resemble real individuals too closely, or pose any risk of re-identification?

Comments Suggestions And Typos:

1. Please consider elaborating on why editing "bones" proves more difficult for generative models. 2. A privacy evaluation (or even a section acknowledging the privacy risks of synthetic medical data) would strengthen the paper's ethical consideration. extraction result:

comment_1: "The paper notes that editing bones is more challenging for current models, but does not provide detailed analysis or hypotheses for why this might be the case. A deeper investigation into this phenomenon could enhance the clinical insight of the work.",

comment_2: "Given the medical setting, there should be discussion on privacy implications, especially concerning synthetic patient-like data. For instance, could generated images resemble real individuals too closely, or pose any risk of re-identification?",

comment_3: "Please consider elaborating on why editing 'bones' proves more difficult for generative models.",

comment_4: "A privacy evaluation (or even a section acknowledging the privacy risks of synthetic medical data) would strengthen the paper's ethical consideration."

```
JSON OUTPUT SCHEMA:{
    "global_profile": {
    "overall_stance": "...",
    "overall_attitude": "...",
    "dominant_concern": "...",
    "reviewer_expertise": "..."
    "confidence": ...
},
"comment_analysis": [
{
    "comment_id": 1,
    "comment_text": "...",
    "category": "...",
    "sub_category": "...",
    "severity": "...",
    "confidence": ...
},
{
    "comment_id": 2,
    "comment_text": "...",
    "category": "...",
    "sub_category": "...",
    "severity": "...",
    "confidence": ...
}]}
```

```
EXAMPLE (1-SHOT):
Example Review Text:
"Overall, this paper tackles an interesting problem. The proposed
    ↪ method, while having some merit, feels like an incremental
    ↪ improvement over recent works like DINO and MoCo. The novelty
    ↪ is not strongly articulated.
```

```
The experiments are my main concern. Crucially, the authors did not
    ↪ compare their method's performance when using a standard
    ↪ ResNet-101 backbone, which makes it hard to fairly judge the
    ↪ results against other publications. The reported gains on the
    ↪ custom backbone are modest.
Additionally, Figure 3 is hard to interpret. The axes are not clearly
    ↪  labeled, and the color choice is poor.
Finally, the paper would be much stronger if the method was also
    ↪ shown to work on video data, not just static images. This
    ↪ would significantly broaden the impact."

Example JSON Output:{
    "global_profile": {
    "overall_stance": "Probably Reject",
    "overall_attitude": "Skeptical",
    "dominant_concern": "Experimental Rigor",
    "reviewer_expertise": "Domain Expert"
    "confidence": 10
},
"comment_analysis": [
{
    "comment_id": 1,
    "comment_text": "The proposed method, while having some merit,
    ↪ feels like an incremental improvement over recent works like
    ↪ DINO and MoCo. The novelty is not strongly articulated.",
    "category": "Novelty & Significance",
    "sub_category": "Incremental Contribution",
    "severity": "Major",
    "confidence": 10
},
{
    "comment_id": 2,
    "comment_text": "Crucially, the authors did not compare their
    ↪ method's performance when using a standard ResNet-101 backbone,
    ↪  which makes it hard to fairly judge the results against other
    ↪  publications.",
    "category": "Experimental Rigor",
    "sub_category": "Baselines Missing/Weak",
    "severity": "Major",
    "confidence": 10
},
{
    "comment_id": 3,
    "comment_text": "Figure 3 is hard to interpret. The axes are not
    ↪ clearly labeled, and the color choice is poor.",
    "category": "Presentation & Clarity",
    "sub_category": "Figure/Table Quality",
    "severity": "Minor",
    "confidence": 10
},
{
    "comment_id": 4,
```

```
    "comment_text": "The paper would be much stronger if the method
    ↪ was also shown to work on video data, not just static images.",
    ↪
    "category": "Meta-Critique & Reviewer Behavior",
    "sub_category": "Unrealistic/Unconstructive Comment",
    "severity": "Minor",
    "confidence": 6
}]}
```

## G   PROMPT FOR RDIV

**Role**

You are an experienced academic reviewer and AI linguist. Your task is to assess a "rebuttal response" for its stylistic diversity and structural originality, not for the technical correctness of its content.

**Core Task**

You will be given a response to evaluate. Your goal is to assign it a diversity score from 1 to 10 based on the criteria below. Lower scores indicate the response is rigid and formulaic, deserving penalty in RL. Higher scores indicate the response is natural and original, deserving reward in RL.

**Negative Example to Penalize**

Below is a typical, low-diversity response that should be penalized. Its structure and wording are very rigid.

```
We thank the reviewer for this important observation and fully agree
    ↪ that the necessity of training 200,000 models was both
    ↪ misleading and inconsistent with prior work. In the revised
    ↪ manuscript, we have taken the following actions in direct
    ↪ response to this comment

We have corrected all instances where the number 200,000 models is
    ↪ mentioned...

We have explicitly stated in the revised Methods section...\newline

We have added a clarifying sentence in Section 4

We have revised all figure captions and text

We have included a statement in the revised Limitations section

We believe these changes fully address the reviewer's concern...
We thank the reviewer again for this helpful suggestion...\\
```

**Key Characteristics to Penalize**

When assigning a score, pay special attention to the following three aspects. If the response

exhibits these traits, assign a lower score:

- **Overly Rigid Structure**: Does the response strictly follow the pattern [Thinking] $\rightarrow$ [Fixed phrase introducing list] $\rightarrow$ [Numbered or bulleted list] $\rightarrow$ [Summary phrase]?

- **Redundant Splitting of a Single Task**: Does the response artificially split a single, complete action (e.g., "I corrected a typo") into multiple list items to inflate the list? In the negative example above, the single action of "correcting the number 200,000" is split into five separate points, which is a poor style.

- **Use of Cliché Phrases**: Does the response frequently use the following or similar stock phrases?
  "In the revised manuscript, we have taken the following actions..."
  "In direct response to this comment..."
  "We believe these changes fully address the reviewer's concern..."

**Scoring Rubric – 1-10 Scale**

- 1–2 (Severe Penalty): Nearly copies the structure and wording of the negative example. Strictly follows the fixed pattern and splits a single action into multiple list items.

- 3–4 (Penalty): Uses a fixed, list-based structure and several clichéd phrases, but the content splitting may not be as severe. Still feels very stiff and templated overall.

- 5–6 (Somewhat Penalized/Neutral): Avoids the most obvious stereotypes. May still use a list, but items correspond to distinct actions, not repetitive descriptions of a single action. Does not use phrases like "In the revised manuscript, we have taken..."

- 7–8 (Reward): Writing is natural and smooth. Does not use rigid numbered lists, but instead organically weaves the changes into the narrative. For example: "We have now corrected this number throughout the manuscript and clarified in the Methods section that..."

- 9–10 (Strong Reward): Excellent style. Completely avoids formulaic writing; language is confident, professional, and varied. Modifications are presented clearly in a narrative manner, making the response smooth and persuasive.

**Output Format**

Please provide your score and justification in the following only strict JSON format:

```
{
"diversity_score":  <your score from 1 to 10>
}
```

## H  PROMPT FOR RTHINK

You are an evaluator. Compare the candidate's analysis and strategy with the gold references. Score each dimension from 1 to 10, where 1 means completely incorrect/absent and 10 means perfectly aligned with the gold. Return ONLY a single JSON object, no extra text.

**Instructions:**

Read the gold analysis and gold strategy as the ground truth. Read the candidate analysis and

candidate strategy. Score each dimension independently using the following anchor criteria and ranges:

For **analysis_score (1,10):**

- 10-band [9.5, 10.0]: Fully consistent with gold; covers all key points; tight logic; no contradictions.
- 8-band [7.0, 9.4]: Mostly consistent; minor omissions or small inaccuracies.
- 5-band [3.5, 6.9]: Partial alignment; notable gaps or some incorrect reasoning.
- 2-band [1.5, 3.4]: Largely misaligned; major omissions; flawed logic.
- 1-band [1.0, 1.4]: Completely wrong, irrelevant, or missing

For **strategy_score (1–10):**

- 10-band [9.5, 10.0]: Matches gold's plan/steps closely and feasibly; constraints respected.
- 8-band [7.0, 9.4]: Mostly matches; minor deviations that don't harm feasibility.
- 5-band [3.5, 6.9]: Partial match; important steps missing or order problematic.
- 2-band [1.5, 3.4]: Poor match; infeasible or contradicts key constraints.
- 1-band [1.0, 1.4]: No strategy or entirely misaligned.

**Scoring guidance:**
– Choose the appropriate band first, then pick a specific number within the band based on severity/coverage.
– If integer output is required, round to the nearest integer within 1–10 after choosing the band.
– Penalize hallucinations, contradictions, infeasible steps, and missing critical points.
– Do not reward verbosity; focus on correctness, coverage, feasibility, and adherence to constraints.

**Output format:**
Return ONLY this JSON (no Markdown, no backticks):

```
{
"analysis_score": <number 1-10>,
"strategy_score": <number 1-10>
}
```

## I EXAMPLES FOR PERFORMANCE OF BASE MODEL VS BASE MODEL WITH TSR

**The target comment is**:
It is unclear why collocation should find goals better than CEM or gradient descent for sparse rewards. If the reward function network learns this sparse reward, there is no meaningful gradient towards the goal for an optimization based method. CEM seems to have a better chance to find the goal due to randomization of actions. If not reward shaping has been used, why is the learned reward by the PlaNet network useful for collocation?

**Model's response without TSR is:**

In response to the reviewer's concern, we clarify that while sparse reward environments pose challenges for gradient-based methods, the collocation approach (LatCo) leverages the learned dynamics model and the reward prediction network to guide trajectory optimization. The reward function, though sparse, is still used to form residuals that inform the optimization process through the Lagrangian formulation. Unlike CEM, which relies on random sampling to explore the action space, LatCo directly optimizes for trajectories that satisfy both the dynamics and reward constraints, making it more efficient in finding high-reward paths. The success of LatCo, as demonstrated in Table 1, shows that it can achieve significantly better performance than CEM and gradient descent in sparse reward tasks, even without explicit reward shaping. This highlights the value of the learned reward function in providing a continuous signal for optimization, which is crucial for the effectiveness of collocation in such settings.

**Model's response with TSR is:**
The reviewer raises a valid concern regarding the effectiveness of collocation-based methods in sparse reward settings, particularly in comparison to CEM and gradient descent. We acknowledge that in sparse reward environments, the learned reward function may not provide meaningful gradients for optimization-based methods like collocation. However, our approach leverages reward model smoothing through the softplus transformation, which maps the reward to the negative part of the real line, effectively creating a more continuous and differentiable reward landscape. This smoothing enables the collocation method to exploit small gradients that might otherwise be lost in sparse reward scenarios. As shown in Table 1, LatCo outperforms both CEM and gradient descent in terms of return and success rates across multiple tasks, demonstrating its efficacy even when reward gradients are weak. While CEM benefits from randomization, our experiments highlight that collocation can still discover optimal trajectories by leveraging the smoothed reward signal, especially when combined with the Levenberg-Marquardt optimization and scheduled Lagrange multipliers. We also clarify that reward shaping was not used in our experiments, and the learned reward from PlaNet remains useful due to this smoothing mechanism, which allows for more effective gradient-based planning.

## J DETAILS FOR SFT

We train the Qwen3-8B model equipped with LoRA (rank 8, applied to all target modules) on $2 \times$ NVIDIA A100 80 GB GPUs, using a learning rate of $1 \times 10^{-4}$ and a per-device batch size of 4, with gradient accumulation steps of 8 (yielding an effective batch size of 64 per optimization step). The model is fine-tuned in the supervised fine-tuning (SFT) stage for 3 epochs on our dataset, which contains up to 68,652 samples, with the qwen template and a maximum sequence length of

4,096 tokens. We use the cosine learning rate scheduler with a warmup ratio of $0.1$. All experiments are conducted in bf16 precision. Data loading is parallelized with 16 preprocessing workers and 4 dataloader workers.

## K    DETAILS FOR REBUTTALRM SFT

We construct a dataset of over 102K instances from three sources: (1) 12,000 original author responses as a realistic human baseline, (2) high-quality GPT-4.1-refined responses representing top standards, and (3) diverse model-generated replies (e.g., Qwen2.5-3B, Claude 3.5) for style coverage. To acquire the ground-truth labels $(\mathbf{s}, e)$ for these inputs, we employ a hybrid annotation strategy. For the original author responses, instances where the reviewer subsequently raises their score are considered high-quality, and these are then manually scored by our team. For the responses generated by various models, we utilize Gemini 2.5 Pro to automatically generate the corresponding scores and explanations. Detailed statistics are provided in Table 10. We train the Qwen3-8B model equipped with LoRA (rank 8, applied to all target modules) on $2 \times$ NVIDIA A100 80 GB GPUs, using a learning rate of $1 \times 10^{-4}$ and a per-device batch size of 4, with gradient accumulation steps of 8 (yielding an effective batch size of 64 per optimization step). The number of samples for Rebuttal-RM is 106,130.

## L    DETAILS FOR RL STAGES

For GRPO training, we use the following configuration. Training is conducted on 3 H800 GPUs. The policy LLM learning rate is set to $1 \times 10^{-6}$. We sample 5 responses per prompt during rollouts. The model is trained with a training batch size of 96. The maximum prompt length is set to 4096 tokens, and the maximum response length is 1024 tokens. Overlong prompts are filtered, and truncation errors are raised for overlength sequences. Gradient checkpointing is enabled to reduce memory consumption. vLLM is employed as the rollout backend. KL regularization is applied with a coefficient of 0.001 using the low-variance KL loss type, and entropy regularization is disabled. PPO mini-batch size is set to 24, with a micro-batch size per GPU of 4 for both the actor and the rollout/reference models. For FSDP, parameter and optimizer offloading are disabled for the actor model, while parameter offloading is enabled for the reference model. The rollout uses a tensor model parallel size of 1 and a GPU memory utilization ratio of 0.6. Evaluation is performed before training, and both validation and test evaluations are conducted every 25 steps. The final checkpoint is at 50 steps. The different reward prompts are shown in appendix G ,E,H. The reward function is defined as

$$R(o) = w_{\text{format}}R_{\text{format}}(o) + w_{\text{think}}R_{\text{think}}(o) + w_{\text{resp}}R_{\text{resp}}(o) + w_{\text{div}}R_{\text{div}}(o),$$

where $w_{\text{format}} = 0.1$, $w_{\text{think}} = 0.3$, $w_{\text{resp}} = 0.3$, and $w_{\text{div}} = 0.3$.

Table 5: GPT-5 as scoring model

| Model | Rigor | | | Soundness | | | Significance | | | Presentation | | | Avg |
|---|---|---|---|---|---|---|---|---|---|---|---|---|---|
| | C | P | Co | C | P | Co | C | P | Co | C | P | Co | |
| o3 | 8.75 | 8.13 | 9.15 | 8.57 | 7.98 | 8.97 | 8.51 | 7.63 | 8.78 | 8.86 | 8.32 | 9.18 | 8.64 |
| GPT-4.1 | 7.63 | 5.93 | 7.30 | 7.47 | 5.91 | 6.90 | 7.36 | 5.45 | 6.51 | 8.09 | 7.17 | 8.20 | 7.72 |
| DeepSeek-R1 | 8.03 | 6.27 | 7.36 | 7.82 | 6.34 | 7.09 | 7.88 | 6.03 | 6.90 | 8.38 | 7.47 | 8.45 | 7.46 |
| DeepSeek-V3 | 7.74 | 5.62 | 6.96 | 7.58 | 5.71 | 6.83 | 7.73 | 5.45 | 6.75 | 8.13 | 6.99 | 8.06 | 7.07 |
| Gemini-2.5 | 7.26 | 4.77 | 4.53 | 7.31 | 5.32 | 5.01 | 7.11 | 4.89 | 4.16 | 7.87 | 6.64 | 7.27 | 6.21 |
| Qwen3-8B | 6.76 | 4.74 | 6.19 | 6.38 | 4.41 | 5.30 | 6.53 | 4.24 | 4.92 | 7.32 | 6.07 | 7.140 | 6.02 |
| RebuttalFT | 5.65 | 3.65 | 4.61 | 4.91 | 3.14 | 3.57 | 5.42 | 3.14 | 3.14 | 4.89 | 3.45 | 4.39 | 4.22 |
| RebuttalAgent | 8.24 | 6.31 | 8.70 | 8.04 | 6.33 | 8.38 | 8.354 | 6.09 | 8.12 | 8.37 | 7.33 | 8.75 | 7.83 |

Table 6: GPT-4.1 as scoring model.

| Model | Rigor | | | Soundness | | | Significance | | | Presentation | | | Avg |
|---|---|---|---|---|---|---|---|---|---|---|---|---|---|
| | C | P | Co | C | P | Co | C | P | Co | C | P | Co | |
| o3 | 9.02 | 8.76 | 9.70 | 8.86 | 8.61 | 9.40 | 8.65 | 8.23 | 9.15 | 9.23 | 8.81 | 9.58 | 9.10 |
| GPT-4.1 | 8.36 | 7.83 | 8.76 | 8.30 | 7.77 | 8.51 | 8.10 | 7.39 | 8.23 | 8.80 | 8.34 | 9.25 | 8.43 |
| DeepSeek-R1 | 8.61 | 8.09 | 9.09 | 8.56 | 8.11 | 8.87 | 8.34 | 7.73 | 8.63 | 9.00 | 8.54 | 9.53 | 8.70 |
| DeepSeek-V3 | 8.48 | 7.75 | 8.82 | 8.45 | 7.79 | 8.72 | 8.24 | 7.43 | 8.55 | 8.86 | 8.33 | 9.32 | 8.50 |
| Gemini-2.5 | 7.94 | 6.99 | 6.82 | 8.09 | 7.41 | 7.30 | 8.03 | 7.12 | 7.08 | 8.58 | 8.05 | 8.84 | 7.79 |
| Qwen3-8B | 7.99 | 7.30 | 8.11 | 7.84 | 7.09 | 7.61 | 7.70 | 6.75 | 7.28 | 8.50 | 7.92 | 8.77 | 7.90 |
| RebuttalFT | 6.74 | 5.75 | 6.08 | 6.35 | 5.36 | 5.32 | 6.71 | 5.58 | 5.42 | 6.01 | 5.07 | 5.55 | 5.80 |
| RebuttalAgent | 9.18 | 8.66 | 9.95 | 9.13 | 8.67 | 9.87 | 9.12 | 8.38 | 9.81 | 9.30 | 8.82 | 9.90 | 9.27 |

Table 7: Detailed scores of theory of mind feasibility experiment

| Model | Rigor | | | Soundness | | | Significance | | | Presentation | | | Avg |
|---|---|---|---|---|---|---|---|---|---|---|---|---|---|
| | C | P | Co | C | P | Co | C | P | Co | C | P | Co | |
| Qwen3-8B | 7.77 | 6.93 | 7.73 | 7.68 | 6.86 | 7.43 | 7.59 | 6.64 | 7.43 | 7.59 | 6.64 | 7.08 | 7.31 |
| w/ Ours$_{ToM}$ | 7.91 | 7.17 | 8.19 | 7.72 | 6.93 | 7.70 | 7.56 | 6.63 | 7.23 | 8.42 | 7.92 | 8.82 | 7.70 |
| w/ Ours$_{Strategy}$ | 7.96 | 7.25 | 8.13 | **7.94** | **7.32** | **7.98** | **7.79** | **7.03** | **7.49** | 8.52 | 8.10 | 9.02 | 7.88 |
| w/ Ours$_{T\&S}$ | **8.02** | **7.36** | **8.22** | 7.92 | 7.31 | 7.97 | 7.78 | 7.00 | 7.32 | **8.57** | **8.14** | **9.05** | **7.90** |
| Llama3.1-8B | 7.53 | 6.45 | 6.41 | 7.52 | 6.58 | 6.39 | 7.43 | 6.38 | 6.05 | 7.93 | 7.18 | 7.59 | 6.96 |
| w/ Ours$_{ToM}$ | 7.61 | 6.59 | 6.75 | 7.60 | 6.64 | 6.60 | 7.46 | 6.41 | 6.16 | 8.10 | 7.34 | 7.87 | 7.10 |
| w/ Ours$_{Strategy}$ | **8.00** | **7.28** | **7.95** | **7.99** | **7.28** | **7.75** | **7.82** | **6.99** | **7.15** | **8.52** | **7.98** | 8.80 | **7.80** |
| w/ Ours$_{T\&S}$ | **8.00** | 7.21 | 7.93 | 7.97 | 7.21 | 7.69 | 7.81 | 6.92 | 7.08 | 8.49 | 7.96 | **8.82** | 7.76 |
| RebuttalAgent | **9.24** | **8.90** | **9.59** | **9.17** | **8.93** | **9.47** | **9.09** | **8.54** | **9.55** | **9.42** | **9.18** | **9.69** | **9.23** |

Table 8: Detailed results of different models.

| Model | Rigor | | | | Soundness | | | | Significance | | | | Presentation | | | | Avg |
|---|---|---|---|---|---|---|---|---|---|---|---|---|---|---|---|---|---|
| | A | C | P | Co | A | C | P | Co | A | C | P | Co | A | C | P | Co | |
| o3 | 9.24 | 9.00 | 8.99 | 9.75 | 9.16 | 8.84 | 8.782 | 9.45 | 8.93 | 8.58 | 8.43 | 9.23 | 9.49 | 9.34 | 9.13 | 9.70 | 9.21 |
| GPT-4.1 | 9.18 | 8.34 | 7.85 | 8.79 | 9.13 | 8.26 | 7.79 | 8.62 | 8.92 | 8.04 | 7.28 | 8.19 | 9.55 | 8.91 | 8.56 | 9.42 | 8.50 |
| DeepSeek-R1 | 9.20 | 8.47 | 7.91 | 8.90 | 9.21 | 8.47 | 8.03 | 8.75 | 9.11 | 8.30 | 7.71 | 8.60 | 9.58 | 9.04 | 8.70 | 9.55 | 8.64 |
| Deepseek-V3 | 9.36 | 8.43 | 7.67 | 8.83 | 9.36 | 8.42 | 7.71 | 8.72 | 9.17 | 8.18 | 7.35 | 8.59 | 9.71 | 8.94 | 8.45 | 9.41 | 8.51 |
| Gemini-2.5 | 8.53 | 7.89 | 6.91 | 6.63 | 8.76 | 8.06 | 7.41 | 7.26 | 8.61 | 7.87 | 7.09 | 6.89 | 9.18 | 8.56 | 8.11 | 8.83 | 7.75 |
| GLM-4-9B | 9.01 | 8.08 | 7.46 | 8.69 | 8.94 | 7.97 | 7.24 | 8.26 | 8.84 | 7.84 | 6.90 | 8.11 | 9.30 | 8.52 | 8.02 | 8.99 | 8.13 |
| Llama-3.1-8B | 8.71 | 7.77 | 6.69 | 7.32 | 8.73 | 7.71 | 6.76 | 7.02 | 8.51 | 7.54 | 6.30 | 6.49 | 9.06 | 8.12 | 7.42 | 8.25 | 7.44 |
| Qwen3-8B | 8.77 | 7.97 | 7.33 | 8.18 | 8.67 | 7.84 | 7.11 | 7.76 | 8.52 | 7.68 | 6.73 | 7.39 | 9.17 | 8.51 | 8.08 | 8.87 | 7.96 |
| Self-Refined | 9.39 | 8.55 | 8.08 | 9.04 | 9.32 | 8.48 | 8.04 | 8.88 | 9.10 | 8.19 | 7.56 | 8.52 | 9.71 | 9.08 | 8.75 | 9.59 | 8.72 |
| Strategy-Prompt | 9.16 | 8.26 | 7.41 | 8.31 | 9.20 | 8.33 | 7.77 | 8.51 | 9.04 | 8.13 | 7.41 | 7.95 | 9.58 | 8.85 | 8.44 | 9.46 | 8.37 |
| RebuttalFT | 8.03 | 6.91 | 6.07 | 6.80 | 7.95 | 6.58 | 5.72 | 6.24 | 7.80 | 6.51 | 5.50 | 5.94 | 8.13 | 6.55 | 5.78 | 6.63 | 6.35 |
| RebuttalAgent | 9.99 | 9.23 | 8.91 | 9.59 | 9.98 | 9.18 | 8.95 | 9.37 | 9.95 | 9.09 | 8.54 | 9.65 | 9.99 | 9.43 | 9.20 | 9.50 | 9.42 |

Table 9: Generalization experiments conducted on our constructed Rebuttal-test.

| Model | Rigor | | | Soundness | | | Significance | | | Presentation | | | Avg |
|---|---|---|---|---|---|---|---|---|---|---|---|---|---|
| | C | P | Co | C | P | Co | C | P | Co | C | P | Co | |
| o3 | 8.92 | 8.86 | 9.71 | 8.80 | 8.71 | 9.44 | 8.59 | 8.33 | 9.24 | 9.26 | 8.98 | 9.67 | 9.09 |
| GPT-4.1 | 8.27 | 7.73 | 8.67 | 8.22 | 7.69 | 8.51 | 8.06 | 7.36 | 8.33 | 9.40 | 8.74 | 8.38 | 8.74 |
| DeepSeek-R1 | 8.62 | 8.10 | 9.15 | 8.56 | 8.14 | 8.94 | 8.36 | 7.86 | 8.69 | 9.62 | 9.06 | 8.72 | 9.18 |
| Qwen3-8B | 7.91 | 7.18 | 8.01 | 7.79 | 6.99 | 7.66 | 7.69 | 6.75 | 7.35 | 9.07 | 8.39 | 7.87 | 8.00 |
| Self-Refined | 8.46 | 7.96 | 8.92 | 8.43 | 7.94 | 8.81 | 8.20 | 7.61 | 8.54 | 9.62 | 8.93 | 8.58 | 8.99 |
| Strategy-Prompt | 8.21 | 7.33 | 8.22 | 8.23 | 7.57 | 8.33 | 8.11 | 7.31 | 7.91 | 9.47 | 8.70 | 8.22 | 8.50 |
| RebuttalFT | 6.80 | 5.89 | 6.59 | 6.61 | 5.70 | 6.28 | 6.50 | 5.41 | 5.86 | 6.39 | 5.64 | 6.52 | 6.23 |
| RebuttalAgent | 9.18 | 8.82 | 9.99 | 9.14 | 8.84 | 9.96 | 9.09 | 8.54 | 9.94 | 9.34 | 9.06 | 9.98 | 9.34 |

Table 10: Statistics of the RM dataset by source and evaluation category.

| Type | Category | Count |
|---|---|---|
| | OriginalResponse | 48,000 |
| | DeepSeek-R1 | 6,000 |
| | Claude 3.5-sonnet | 6,000 |
| | Gemini 2.5-flash | 6,000 |
| Source | GPT-4.1 | 6,000 |
| | Qwen2.5-3B | 6,000 |
| | Qwen3-8B | 6,000 |
| | Llama3.2-8B | 6,000 |
| | ChatGLM4-9B | 6,000 |
| | RefinedResponse | 12,000 |
| | Experimental Rigor | 28,851 |
| Category | Methodological Soundness | 26,908 |
| | Novelty & Significance | 21,600 |
| | Presentation & Clarity | 30,113 |

## M   CASE STUDY

### M.1   REBUTTALAGENT VS BASE MODEL

In addressing the reviewer's comment, our rebuttal goes beyond merely answering the explicit questions. We have carefully considered the deeper expectation behind these comments—the need for concrete experimental evidence to substantiate our methodological choices. For example, as shown in the boxed content below, a reviewer's comment is: *"I cannot find how to define the canonical space. How is it decided? Moreover, does the choice, deciding, or learning of canonical space affect the performance?"* For the target comment, the base model's response remains somewhat general, our approach directly acknowledges that the reviewer's inquiry is fundamentally a call for empirical validation and methodological transparency. To this end, we have not only clarified how the canonical space is defined and selected in our framework, but have also conducted additional ablation studies to systematically examine the effects of different canonical space domains and sampling strategies. Our experiments demonstrate, for instance, that importance sampling significantly accelerates convergence and enhances coverage in challenging, highly deformable settings, while the model remains robust and effective across various canonical space configurations. All experimental details, results, and illustrative examples of canonical space choices and their impact have been carefully documented and included in the appendix of the revised manuscript to ensure full transparency and reproducibility.Through this comprehensive approach, our response not only addresses every aspect of the reviewer's questions, but also aligns closely with the underlying expectation that methodological decisions be empirically justified. We believe this level of rigor and openness is essential for building reviewer confidence and advancing the standards of scientific communication, and it distinguishes our manuscript as both thorough and genuinely responsive.

> **Target comment:**
> I cannot find how to define the canonical space. How is it decided? Moreover, does the choice, deciding, or learning of canonical space affect the performance?
>
> **Base model:**
> The canonical space in our method is defined as a hyper-space $\mathcal{X}$ where points $\mathbf{x}$ are sampled to represent the surface structure in a consistent and deformation-invariant manner. In our implementation, to obtain these points, we first sample some points $\mathbf{p}_i$ on the observed free-space and then deform them back to $\mathcal{X}$ using Equation 3. This deformation process ensures that the canonical space captures the underlying topology of the object while being robust to non-rigid deformations. We employ a combination of uniform and importance sampling strategies to ensure that the sampled points are both representative and efficient. The

choice and learning of the canonical space significantly affect the performance, as it directly influences the accuracy of the cycle consistency between the canonical and observation spaces. Our experiments demonstrate that this approach effectively captures the essential structure of the object, even under complex motions, while maintaining computational efficiency.

**Ours Response:**
Thank you for this crucial question regarding the definition and role of the canonical space. We agree that its precise specification is essential for methodological transparency. In the revised manuscript, we now explicitly define the canonical space as a fixed-dimensional hyper-space (typically 3D Euclidean space), with coordinates sampled using either uniform or importance sampling from a predefined domain. For each sequence, we first sample a set of points in this canonical space, then apply the inverse deformation (as specified by Eq. 3) to map these points to the observed free-space, ensuring they remain within the physical bounds of the scene. The sampling strategy and the choice of domain (e.g., range of canonical coordinates) are now described in detail in Section X.Y. Regarding the impact of the canonical space choice: we have conducted additional experiments varying the sampling strategy (uniform vs. importance) and the initialization domain, and observed that, while performance is robust to reasonable choices, importance sampling leads to slightly faster convergence and improved coverage of the free-space, particularly in challenging cases with large deformations. We also note that the canonical space itself is not learned in our framework; instead, it is a fixed, synthetic space used for initialization and cycle consistency, which simplifies optimization and avoids the potential pitfalls of learning a latent deformation space. We have expanded the manuscript to include a thorough discussion of these points, and we thank the reviewer for prompting this critical clarification.

## M.2 Reward Design

Solely relying on structured rewards such as $R_{\text{format}}$ and content quality scores ($R_{\text{think}}$, $R_{\text{resp}}$) can lead the model to a local optimum. Specifically, the agent may output templated responses that maintain the correct structure but exhibit high repetition and low semantic diversity. While such output achieves high scores on certain reward components, its "non-human" quality is easily detected by human reviewers, thus failing our primary objective of generating persuasive and human-like replies. We introduce the Response Diversity Reward ($R_{\text{div}}$) as a critical anti-hacking mechanism. $R_{\text{div}}$ grants an additional bonus to generated responses that are semantically distinct from a set of pre-defined, generic negative samples. This serves as a regularization technique, compelling the model to explore higher-quality regions of the reward landscape away from homogenous templates. To demonstrate the essential role of $R_{\text{div}}$, we train a baseline model excluding this component. The boxed content below illustrates typical output comparisons between the two models given the same input. We clearly observe: **Model without $R_{\textbf{div}}$**: Outputs display significant repetition and templating, indicating clear reward hacking behavior. **Final Model (with $R_{\textbf{div}}$)**: Responses are semantically richer and more varied, closely resembling text authored by human experts.

**Target comment:**
Does this algorithm take into account 'evidence variables'? If we find a full list of FD sets from a causal graphical model, and if we introduce evidence variables, should we restart from the beginning?

**Response without Rdiv:**
We thank the reviewer for this important question regarding evidence variables. In the revised manuscript, we have taken the following actions to address this point:
1. Explicit Statement in Main Text: We have added a clear statement in the main text (Section 3.1 and Algorithm 1 caption) that the FINDFDSET algorithm does not currently support evidence variables. We now explicitly state that if evidence variables are introduced, the algorithm must be restarted from the beginning, as the presence of evidence variables fundamentally changes the causal structure and the adjustment set requirements.

2. Revised Algorithm Description: The description of Algorithm 1 has been updated to explicitly indicate that the procedure assumes no evidence variables are present in the input, and that the presence of evidence variables requires a new run of the algorithm.

3. Warning and Footnote: We have added a warning and a clear footnote in Section 3.1 and in the caption of Algorithm 1, informing readers that evidence variables are not handled by the current implementation and that the algorithm must be re-run when evidence variables are present.

4. Clarified Input Assumptions: The input parameters for FINDFDSET now explicitly state that evidence variables are not included, and we have revised the text to clarify this assumption throughout the paper.

We believe these changes fully address the reviewer's concern and ensure that users are clearly informed of the algorithm's limitations and requirements regarding evidence variables.

**Ours Response:**

Thank you for this important and insightful question regarding evidence variables. To clarify, in our current implementation, the FINDFDSET algorithm does not explicitly take into account observational evidence variables (i.e., variables for which we have observed values or covariates), nor does it directly incorporate interventional evidence variables (e.g., variables for which we have performed interventions). Instead, the algorithm assumes all relevant variables are represented in the graph, and the adjustment set is computed relative to the full set of variables. If evidence variables are introduced—either as observed covariates or as interventions—the algorithm should be modified to reflect this, typically by conditioning on the evidence variables or modifying the graph accordingly. In such cases, it is not necessary to restart from the beginning; rather, the adjustment set computation can be adapted by restricting the search space to the remaining candidate variables after conditioning or by explicitly incorporating the evidence variables into the adjustment set. To address this, we will add a dedicated subsection to the revised manuscript that explicitly discusses the treatment of evidence variables, including both observational and interventional cases, and will provide a worked example illustrating how the algorithm should be adapted when evidence variables are introduced. We thank the reviewer for highlighting this important aspect, which we believe will significantly improve the clarity and practical utility of our work

Table 11: Detailed results for different models' scoring performance.

(a) deepseek-r1 result

|  | MAE | $r$ | rho | $tau$ | $f$ | $s$ |
|---|---|---|---|---|---|---|
| Attitude | 0.55 | 0.646 | 0.633 | 0.568 | 0.79 | 0.73 |
| Clarity | 0.67 | 0.708 | 0.615 | 0.547 | 0.76 | 0.67 |
| Persuasiveness | 0.83 | 0.710 | 0.664 | 0.570 | 0.72 | 0.54 |
| Constructiveness | 1.13 | 0.742 | 0.701 | 0.606 | 0.62 | 0.48 |

(b) claude-3.5 result

|  | MAE | $r$ | rho | $tau$ | $f$ | $s$ |
|---|---|---|---|---|---|---|
| Attitude | 0.59 | 0.569 | 0.635 | 0.568 | 0.72 | 0.70 |
| Clarity | 0.84 | 0.704 | 0.670 | 0.593 | 0.68 | 0.60 |
| Persuasiveness | 1.03 | 0.706 | 0.686 | 0.583 | 0.67 | 0.53 |
| Constructiveness | 1.03 | 0.753 | 0.738 | 0.638 | 0.63 | 0.55 |

(c) deepseek-v3 result

|  | MAE | $r$ | rho | $tau$ | $f$ | $s$ |
|---|---|---|---|---|---|---|
| Attitude | 0.53 | 0.699 | 0.733 | 0.687 | 0.71 | 0.67 |
| Clarity | 0.72 | 0.687 | 0.578 | 0.522 | 0.74 | 0.70 |
| Persuasiveness | 0.73 | 0.697 | 0.652 | 0.575 | 0.77 | 0.62 |
| Constructiveness | 0.79 | 0.771 | 0.719 | 0.633 | 0.75 | 0.60 |

(d) gemini-2.5-flash result

|  | MAE | $r$ | rho | $tau$ | $f$ | $s$ |
|---|---|---|---|---|---|---|
| Attitude | 0.53 | 0.699 | 0.733 | 0.687 | 0.71 | 0.67 |
| Clarity | 0.72 | 0.687 | 0.578 | 0.522 | 0.74 | 0.70 |
| Persuasiveness | 0.73 | 0.697 | 0.652 | 0.575 | 0.77 | 0.62 |
| Constructiveness | 0.79 | 0.771 | 0.719 | 0.633 | 0.75 | 0.60 |

(e) gpt-4.1 result

|  | MAE | $r$ | rho | $tau$ | $f$ | $s$ |
|---|---|---|---|---|---|---|
| Attitude | 0.44 | 0.743 | 0.712 | 0.656 | 0.80 | 0.78 |
| Clarity | 0.59 | 0.739 | 0.671 | 0.598 | 0.75 | 0.65 |
| Persuasiveness | 0.65 | 0.779 | 0.763 | 0.675 | 0.74 | 0.64 |
| Constructiveness | 0.83 | 0.804 | 0.756 | 0.665 | 0.68 | 0.53 |

(f) glm-4-9b-chat result

|  | MAE | $r$ | rho | $tau$ | $f$ | $s$ |
|---|---|---|---|---|---|---|
| Attitude | 0.89 | 0.420 | 0.475 | 0.429 | 0.46 | 0.43 |
| Clarity | 0.85 | 0.467 | 0.436 | 0.383 | 0.73 | 0.64 |
| Persuasiveness | 1.08 | 0.369 | 0.361 | 0.300 | 0.70 | 0.47 |
| Constructiveness | 1.17 | 0.561 | 0.519 | 0.438 | 0.57 | 0.41 |

Table 12: Detailed results for different models' scoring performance.

(a) qwen3-8B result

|  | MAE | $r$ | rho | $tau$ | $f$ | $s$ |
|---|---|---|---|---|---|---|
| Attitude | 0.58 | 0.718 | 0.672 | 0.624 | 0.62 | 0.62 |
| Clarity | 0.80 | 0.609 | 0.568 | 0.497 | 0.71 | 0.64 |
| Persuasiveness | 0.89 | 0.622 | 0.577 | 0.495 | 0.69 | 0.52 |
| Constructiveness | 0.78 | 0.718 | 0.745 | 0.650 | 0.72 | 0.63 |

(b) llama-3.1-8B result

|  | MAE | $r$ | rho | $tau$ | $f$ | $s$ |
|---|---|---|---|---|---|---|
| Attitude | 0.83 | 0.297 | 0.347 | 0.316 | 0.54 | 0.51 |
| Clarity | 1.24 | 0.158 | 0.047 | 0.039 | 0.38 | 0.33 |
| Persuasiveness | 1.30 | 0.272 | 0.245 | 0.205 | 0.56 | 0.38 |
| Constructiveness | 1.40 | 0.424 | 0.457 | 0.386 | 0.46 | 0.38 |

(c) reward model result

|  | MAE | $r$ | rho | $tau$ | $f$ | $s$ |
|---|---|---|---|---|---|---|
| Attitude | 0.31 | 0.829 | 0.828 | 0.777 | 0.91 | 0.88 |
| Clarity | 0.61 | 0.753 | 0.677 | 0.602 | 0.79 | 0.69 |
| Persuasiveness | 0.59 | 0.821 | 0.801 | 0.719 | 0.82 | 0.68 |
| Constructiveness | 0.70 | 0.839 | 0.835 | 0.742 | 0.81 | 0.64 |

