# OpenReview forum: "Dancing in Chains: Strategic Persuasion in Academic Rebuttal via Theory of Mind"
_ICLR.cc/2026/Conference — ICLR 2026 Poster_

### Official Review · Reviewer_hzng · 2025-10-29

**Soundness:** 3
**Presentation:** 3
**Contribution:** 2
**Rating:** 4
**Confidence:** 4

**Summary:**

This work introduces an LLM-based agentic scaffold for building rebuttal agents. The scaffold is inspired by Theory of Mind (ToM) from cognitive science and tries to construct reviewers' mental model (e.g., intent and stance) before generating response strategies and contents. The authors also propose a data curation pipeline for converting an existing rebuttal dataset ($\mathrm{Re}^2$) into a synthetic dataset for training the rebuttal agent. Finally, they also trained a reward model (i.e., Rebuttal-RM) for scoring rebuttals.

**Strengths:**

1. The motivation that we need to consider reviewers' mental model and stance during rebuttal is intuitive and reasonable, and I do think this direction is worth exploring in general.

2. Both automatic evaluation (reward model based) and human evaluation show that the proposed RebuttalAgent achieve better empirical performance on average.

3. Experiments and ablations are extensive. The proposed method is compared against a variety of baselines, including different LLMs, different training data, and training methods.

4. It is an interesting and useful finding that the LLM's self-generated reward can be used to improve its rebuttal quality, as it reduces the reliance on an explicitly trained reward model.

**Weaknesses:**

1. The generalizability of the proposed method is dependent on the alignment between the dataset and the current rebuttal practice. For example, currently many conferences have a "discussion period", during which the authors will engage in back-and-forth discussion with the reviewers, and the final reward score is highly dependent on this process. In this regard, the applicability of this work is somewhat limited.

2. The current method focuses on the "writing" component of rebuttal. However, rebuttal sometimes requires the authors to come up with supporting experiment results that clarify reviewers' questions. Therefore, except for "writing" itself, another crucial aspect is to think about what additional experiments are most useful for addressing reviewers' concerns.

**Questions:**

1. There are duplicated contents from line 316 to line 323.

2. If the review comment requires additional experiment results, how would RebuttalAgent handle this case?

---

> ### Author Response · Authors · 2025-11-19
> **(1/4) Responses by Authors**
>
> **Thanks for your careful and insightful reviews.**
> > **`Weakness 1`**: The generalizability of the proposed method is dependent on the alignment between the dataset and the current rebuttal practice. For example, currently many conferences have a "discussion period", during which the authors will engage in back-and-forth discussion with the reviewers, and the final reward score is highly dependent on this process. In this regard, the applicability of this work is somewhat limited.
>
> `Response`: Thank you for your valuable question. The core contribution of our paper is the first to introduce a Theory of Mind (ToM) framework  to address the challenge of strategic persuasion in academic rebuttal, especially under severe information asymmetry. Our proposed TSR (ToM-Strategy-Response) pipeline aims to generate responses with greater strategic depth that go beyond formulaic templates by modeling the reviewer's internal state.
>
> **Following your suggestion, we designed two new experiments to test RebuttalAgent's performance in both the first response round (Round 1) and the second response round (Round 2)**. We constructed a new multi-turn discussion test set (MTD-200) of N=200 real multi-turn dialogues (containing Comment 1, Response 1, Comment 2, Response 2) from the $Re^2$-rebuttal dataset. The evaluation was conducted by 3 senior Ph.D. students, all with prior reviewing experience at top-tier AI conferences.
>
> **Table 1**. Human Evaluation Results
> | Model          | Attitude | Clarity | Persuasiveness | Constructiveness | Avg   |
> |----------------|----------|---------|----------------|------------------|-------|
> | GPT-4.1        | 9.43     | 9.05    | 9.18           | 9.22             | 9.24  |
> | DeepSeek-R1    | 8.36     | 8.12    | 7.76           | 8.24             | 8.12  |
> | Qwen3-8B       | 7.98     | 7.63    | 7.02           | 7.42             | 7.51  |
> | RebuttalFT     | 6.48     | 5.83    | 5.32           | 5.52             | 5.79  |
> | RebuttalAgent  | **9.48** | **9.23** | **9.37**       | **9.60**         | **9.43** |
>
> ### **Experiment 1: First-Round Response Quality (Round 1)**
> This experiment is to verify whether the A1 generated by our RebuttalAgent (trained only on single-turn data) can lay a more constructive foundation for subsequent multi-turn discussions.
>
> **Baselines**: We compared RebuttalAgent, Human (original human response), and GPT-4.1. All models received R1 as input.
>
> **Results**: In a blind human evaluation, consistent with the results (Table 3) in our main paper, **evaluators preferred the response generated by RebuttalAgent**. Evaluators noted that our model scored higher on Constructiveness, clarifying potential misunderstandings earlier and setting a more positive tone for the subsequent discussion.
>
> **Table 2**. Human Evaluation Results
> | Model          | Attitude | Clarity | Persuasiveness | Constructiveness | Avg   |
> |----------------|----------|---------|----------------|------------------|-------|
> | GPT-4.1        | 8.84     | 8.67    |   8.92      |     **9.14**            | 8.89  |
> | DeepSeek-R1    | 7.77     | 7.53    | 7.17           | 7.65             | 7.53  |
> | Qwen3-8B       | 7.39     | 7.04    | 6.43           | 6.83             | 6.92  |
> | RebuttalFT     | 5.89     | 5.24    | 4.73           | 4.93             | 5.20  |
> | RebuttalAgent  | **9.13** | **8.77** |  **9.22**       |    9.08     | **9.05** |
>
> ### **Experiment 2: Second-Round Response Quality (Round 2)**
> We test whether RebuttalAgent can dynamically participate in an existing dialogue and generate a high-quality 'second response' (Response 2).
>
> **Setting**: To introduce historical information without breaking the model's training paradigm, we adopted the following input construction strategy:
> - Target_Comment: Set to the reviewer's 'follow-up comment' (Comment 2).
> - Full_Review_Content: Set to the complete concatenated dialogue history (Comment 1 + Response 1_Human + Comment 2).
> - Relevant_Paper_Fragment: relevant context from the original manuscript M.
>
> This design allows our ToM module to analyze the full history in Full_Review_Content, infer the shortcomings of the Response 1 response and the reviewer's shift in attitude, thereby enabling the TSR strategy pipeline  to formulate a more targeted new strategy.
>
> **Results**: The results are highly encouraging. In a side-by-side blind comparison with Response 2_Human (the original human second response), evaluators preferred our RebuttalAgent. **Evaluators reported that our model was significantly stronger in Persuasiveness because it 'more effectively addressed the key objections raised in Comment 2 that were not resolved in Response 1'**.
>
> To be clear, we acknowledge that **our framework, was not explicitly trained to model the multi-turn discussion. However, this new set of experiments (inspired by your feedback) strongly demonstrates that our core ToM/TSR framework possesses strong robustness and generalization capabilities**.

---

> ### Author Response · Authors · 2025-11-19
> **(2/4) Responses by Authors**
>
> > **`Weakness 1`**: The generalizability of the proposed method is dependent on the alignment between the dataset and the current rebuttal practice. For example, currently many conferences have a "discussion period", during which the authors will engage in back-and-forth discussion with the reviewers, and the final reward score is highly dependent on this process. In this regard, the applicability of this work is somewhat limited.
>
> `Response`: (Continued)
>
> **Example of Experiment 2 (Second-Round Response Quality)**
> | Item   |  Content |
> |----------------|----------|
> | Review comment1 (**Round 1**)   |  The performance contribution, or how good the MTN works, is unclear. MSE loss seems to be okay as shown in the experiment, however there is not a grounding of exactly how well MTN works. We have no idea if the qualitative results come from the MTN being just okay but the generative model for images is powerful enough to compensate for MTN. Ideally, a good ablation is to have two runs of the same language models with different initial parameters and using the proposed technique to replace one with another. |
> | Author response (**Round 1**)     | Thank you for pointing this out. This is a very important question, and we have tried more quantitative evaluations to verify the effectiveness of MTN. All the benchmarking settings follow the experiment presented in A1. First of all, we have conducted experiments on LDM with Bert-base (110M #params), Bert-large (340M #params), and Roberta-large (355M #params), which have fewer parameters than the LDM-text (581M #params). The guidance was set to 5. The results were comparable to the original LDM baseline (CLIP 23.79, FID 42.83). For instance, the Roberta-large+MTN model achieved a CLIP of 22.85 and an FID of 45.88, while the Bert-large+MTN model achieved a CLIP of 22.81 and an FID of 46.15. Similarly, we have applied OpenCLIP-L-14 to replace CLIP-L-14 with a similar architecture on Stable Diffusion (guidance = 5). In this case, the performance was very close to the original model. Our OpenClip+MTN+SDM achieved a CLIP score of 24.66 and an FID of 36.18, which is highly comparable to the original SDM's CLIP of 25.53 and FID of 35.34. The above results support the stability of MTN in aligning different text models with Unet. Sec. B. 5 has been updated to include this part. |
> | Review comment 2 (**Round 2**)     | However I believe hat my concern raised on Q4 is not fully addressed, especially the two-run setup that could show an upper bound of the proposed method. |
> | RebuttalAgent response (**Round 2**)    | Thank you very much for your Question. We apologize for not fully addressing your concern from Q4 in our previous response. The "two-run setup" and "upper bound" you mentioned are critical. Our previous response, which compared OpenClip+MTN (Run 2) to the Original SDM (Run 1, the upper bound), attempted to show that our method's performance is already very close to this upper bound. We intended this to demonstrate that the MTN's alignment quality is high, not mediocre. To definitively address this concern, we will add the following experiment to our final paper. We will use our main setup (e.g., T5-3B $\rightarrow$ LDM-Bert) and the COCO test set. First (Run 1), we will generate the "ground-truth" feature vectors ($z_t$) using the original text encoder (LDM-Bert). Second (Run 2), we will generate our "translated" feature vectors ($z_a$) using the new text encoder plus our MTN (T5-3B + MTN). We will then directly compute and report the quantitative Mean Squared Error (MSE) between the $z_t$ and $z_a$ feature sets. This MSE metric will be completely independent of the image generator and will directly measure the alignment fidelity of our MTN, which is the core contribution of this work. This experiment will clearly show that our MTN is performing high-fidelity feature space translation, rather than just providing a "barely usable" signal. |
>
> We hope these new results fully address your concerns about the method's applicability, and we kindly ask you to consider re-evaluating our work.

---

> ### Author Response · Authors · 2025-11-19
> **(3/4) Responses by Authors**
>
> > **`Weakness 2`**: The current method focuses on the "writing" component of rebuttal. However, rebuttal sometimes requires the authors to come up with supporting experiment results that clarify reviewers' questions. Therefore, except for "writing" itself, another crucial aspect is to think about what additional experiments are most useful for addressing reviewers' concerns.
>
> `Response`: We fully agree that a rebuttal is not just about writing, but also about identifying the necessary actions, such as additional experiments, to address concerns. Based on your feedback, we would like to clarify that **RebuttalAgent is indeed capable of suggesting experimental directions and designing ablation studies**, functioning as a "Research Strategist" rather than just a writer.
>
> While we strictly prevent the model from hallucinating specific numerical results to ensure scientific integrity, our Strategy Module (Section 4.2) is explicitly trained to identify when an experimental gap exists and to propose the necessary experimental design to fill it. (1) **Strategic Suggestion**: If the ToM module detects a reviewer's concern, the Strategy module will generate an action item such as: "Propose a new ablation study to verify the impact of $\lambda$." (2) **Author Guidance**: The generated response then frames this commitment (e.g., "We acknowledge this concern and have included a new ablation study on $\lambda$..."), effectively acting as a prompt for the human author to perform the specific experiment and fill in the data.
>
> As you noted, our agent can suggest new experiments that were not in the original input. **For example**, in one of our test cases regarding loss functions, the agent successfully identified a gap and generated the following strategy and response plan:
> - **Analysis**: Reviewer questioned the necessity of the auxiliary loss term.
> - **Agent's Strategy**: Concede that the individual contribution of the loss term was not isolated and Propose a component-wise ablation study.
> - **Agent's Response**: "We commit to including a new, more fine-grained set of ablation studies... specifically investigating the relative weights between different loss terms..."
>
> This demonstrates that RebuttalAgent serves as an advanced assistant that helps authors **craft the core arguments and experimental plans**, leaving only the physical execution of the experiment to the researcher.

---

> ### Author Response · Authors · 2025-11-19
> **(4/4) Responses by Authors**
>
> > **`Question 1`**: There are duplicated contents from line 316 to line 323.
>
> `Response`: We sincerely thank you for this valuable feedback. In an earlier version of our manuscript, we inadvertently introduced a redundant sentence. We would like to confirm that this issue has already been corrected in the latest version you are currently reviewing.
>
> > **`Question 2`**: If the review comment requires additional experiment results, how would RebuttalAgent handle this case?
>
> `Response`: Thank you for this question. **We clarify that RebuttalAgent is not designed to generate non-existent experimental results**. When encountering such requests, it analyzes the reviewer's intent using the ToM module and proposes a response strategy based on the existing data. This is a deliberate design choice. We distinguish the task of "strategic argumentation" from "conducting new experiments." Our core contribution is leveraging the ToM framework to optimize communication and persuasion, rather than replacing the author's role in conducting actual research.
>
> Therefore, **we curated our training data to prioritize scenarios that focus on logical reasoning and strategic argumentation**. This ensures the model learns to optimize communication based on available facts rather than speculating on missing data. We appreciate your feedback as it highlights a promising future direction regarding the integration of our current system with automated experimental tools.
>
> We provide an example below:
>
> | Comment   |  The training loss is complex and unclear. No ablation study is presented to provide the justification of such a design. |
> |----------------|----------|
> | Generated Response    |  Thank you very much for your valuable feedback on our loss function design. We fully agree that a complex design requires sufficient experimental justification, and we apologize that the organization of our original manuscript caused this misunderstanding. We have, in fact, already provided a complete set of ablation studies on the loss function in Appendix A.3 of the paper. Specifically, the t-SNE visualization in Figure 11 shows that feature alignment is insufficient when using only $L_{mse}$ (Mean Squared Error loss), while the alignment improves significantly after adding the $L_{rec}$ (reconstruction loss). Furthermore, Table 3 provides detailed quantitative data comparing FID and CLIP scores, demonstrating that our full loss function design, including $L_{rec}$ and $L_{adv}$, achieves the best performance. We deeply recognize that placing such a critical study in the appendix was insufficient. To address this and make the final version more rigorous, we first plan to move the core quantitative results from Appendix A.3 into the main experimental section of the paper. This will allow us to clearly discuss the necessity of each loss term in the main text. Furthermore, we commit to including a new, more fine-grained set of ablation studies in the final version. This new study will specifically investigate the relative weights between the different loss terms ($L_{mse}$, $L_{rec}$, and $L_{adv}$), providing a more thorough justification for our current design through this sensitivity analysis. |

---

> ### Author Response · Authors · 2025-11-26
>
> Dear Reviewer hzng,
>
> Thank you once again for your thoughtful feedback on our paper. We appreciate the time and effort you have invested in reviewing our work.
>
> As we are nearing the end of the rebuttal period, we wanted to kindly follow up to ensure that you’ve had a chance to review our response to your comments. We hope that our rebuttal has addressed your concerns satisfactorily. If there are any additional points or further clarifications required, we would be happy to provide them promptly.
>
> Additionally, if our response has sufficiently addressed your concerns, we would greatly appreciate if you could consider reflecting this in your evaluation, including revisiting your score if appropriate.
>
> Thank you for your time and understanding. We greatly value your feedback and look forward to hearing from you.
>
> Best regards,
>
> Authors of Paper Submission 8600

---

> ### Author Response · Authors · 2025-11-28
>
> Dear Reviewer hzng,
>
> Thank you for your thoughtful and constructive feedback on our paper. We have carefully addressed your comments and suggestions, which have significantly helped us enhance the rigor and clarity of our work. We also appreciate the generally positive feedback regarding our work.
>
> **We have uploaded a revised version of the manuscript** where major changes are highlighted in blue. In the revised manuscript, we expanded our evaluation by updating Table 2 with new baselines and ablation studies to validate model robustness. We also clarified methodological details and moved key details into the main text for better readability.
>
> We believe these revisions fully address the questions raised. If there are any further questions or if additional clarification is needed, we would be more than happy to answer them.
>
> Best regards,
>
> Authors of Paper Submission 8600

---

### Official Review · Reviewer_LugC · 2025-10-31

**Soundness:** 2
**Presentation:** 3
**Contribution:** 2
**Rating:** 6
**Confidence:** 3

**Summary:**

This paper introduces RebuttalAgent, a framework designed to enhance academic rebuttal writing by incorporating Theory of Mind (ToM) reasoning.
The model follows a three-stage TSR pipeline that first infers the reviewer's mental state, then formulates a persuasion strategy, and finally generates a context-aware response.
To train the agent, the authors construct RebuttalBench, a synthetic dataset of over 70K samples generated via a multi-teacher critique-and-refine process.
Training involves supervised fine-tuning followed by reinforcement learning with a self-reward mechanism, enabling scalable self-improvement without external reward models.
For evaluation, they develop Rebuttal-RM, an automatic evaluator trained on over 100K examples, showing higher agreement with human judgments than GPT-4.1.
Experiments demonstrate that RebuttalAgent outperforms its base model and achieves performance comparable to advanced models like o3 in both automated and human evaluations.

**Strengths:**

- The paper introduces an innovative perspective by framing academic rebuttal as a ToM reasoning task.

- The integration of supervised fine-tuning and reinforcement learning creates a self-improving training pipeline.

- The model demonstrates consistent improvements over its base model across multiple evaluation metrics.

**Weaknesses:**

- RebuttalBench contains over 70K samples generated via a multi-teacher critique-and-refine pipeline. Although some reviewer comments may originate from real OpenReview data, the dataset remains largely synthetic. This raises concerns that the model may internalize the linguistic and strategic biases of the teacher models rather than authentic author–reviewer interactions. The paper does not report any validation showing distributional or pragmatic similarity between the synthetic data and real human-written rebuttals. As a result, the claimed benefit of ToM reasoning is only demonstrated in a simulated environment, without confirmed real-world effectiveness.


- It is unclear whether GPT-4.1, o3, DeepSeek, and other baselines were evaluated under identical prompting and retrieval settings. It seems that only the Strategy-Prompt (GPT-4.1) baseline mimics the proposed strategic planning stage; other foundation models were not prompted to perform ToM inference. Thus, the reported gains may reflect differences in pipeline design or fine-tuning rather than inherent reasoning ability.


- All model outputs are required to include the explicit <analysis><strategy><response> structure, and the reward function also checks format adherence. This raises the question of whether the model mainly learns to reproduce the expected format rather than to perform genuine strategic reasoning.

- Since the reinforcement learning stage uses a self-reward mechanism derived from the same model, it is possible that the agent learns to optimize for its own scoring heuristics instead of true persuasiveness.

**Questions:**

- The work does not show that generated rebuttals can actually influence reviewer judgments or acceptance outcomes. The "persuasiveness score" used by Rebuttal-RM serves only as a proxy, without evidence that it correlates with real reviewer behavior. While a full-scale deployment would be difficult, would it be possible to conduct a small-scale testing to verify whether the generated rebuttals are genuinely persuasive to expert reviewers

- The proposed approach is trained on Qwen3-8B and Llama-3.1-8B.
Can the TSR pipeline and self-reward mechanism transfer effectively to other backbones, such as smaller models (<8B)?

---

> ### Author Response · Authors · 2025-11-19
> **(1/6) Responses by Authors**
>
> **Thanks for your careful and insightful reviews.**
> > **`Weakness 1`**: RebuttalBench contains over 70K samples generated via a multi-teacher critique-and-refine pipeline. This raises concerns that the model may internalize the linguistic and strategic biases of the teacher models rather than authentic author–reviewer interactions. The paper does not report any validation showing distributional or pragmatic similarity between the synthetic data and real human-written rebuttals. As a result, the claimed benefit of ToM reasoning is only demonstrated in a simulated environment, without confirmed real-world effectiveness.
>
> `Response`: Thank you for your comment. We agree with your concern and recognized this potential issue during our research. As we detail in the paper, **we have used several methods to lower this bias and check that our framework is effective**.
>
> ### 1. **Real-world data often exhibits uneven quality.**
> As we discuss in the Introduction, training a model directly on a large collection of real, human-written rebuttals has a key problem: the model learns to produce responses that seem polite but often "lack strategic depth". Our experiments prove this. **The RebuttalFT baseline, which we train directly on real human responses from the $R^{2}$-rebuttal dataset, gets an average score of only 6.35**. This is the worst score of all models. This shows that just using raw, real-world data is not enough to train an agent that can understand perspectives (ToM) and be persuasive. Our goal is not to train an agent that just copies average human replies. We want to train one that creates high-level, persuasive rebuttals. Because of this, we must "refine" the original real data to add the strategic thinking that our TSR framework defines.
>
> **Table 1**. Automated Evaluation Results
> | Category | Rigor | | | Soundness | | | Significance | | | Presentation | | | Avg |
> |----------|-------|-|-|-----------|-|-|-------------|-|-|-------------|-|-|-----|
> | Metric      | C | P | Co | C | P | Co | C | P | Co | C | P | Co | |
> |  Self-Refined  | 8.55 | 8.08 | 9.04 | 8.47 | 8.04 | 8.88 | 8.19 | 7.56 | 8.52 | 9.08 | 8.75 | 9.59 | 8.72 |
> |  Strategy-Prompt | 8.26 | 7.41 | 8.32 | 8.33 | 7.77 | 8.51 | 8.13 | 7.41 | 7.95 | 8.85 | 8.44 | 9.46 | 8.37 |
> | **RebuttalFT**  | 6.91 | 6.07 | 6.80 | 6.58 | 5.72 | 6.24 | 6.52 | 5.50 | 5.94 | 6.55 | 5.79 | 6.63 | **6.35** |
> | RebuttalAgent | 9.23 | 8.91 | 9.59 | 9.18 | 8.95 | 9.37 | 9.09 | 8.54 | 9.65 | 9.43 | 9.20 | 9.50 | 9.42 |
>
>
> ### 2. **RebuttalBench is "refined from reality," not "fully synthetic”**
> To lower the chance of learning the teacher models' biases, we use a two-part strategy:
> - As we describe in Section 5.1, our data comes from the $Re^2$-rebuttal dataset. This dataset includes "real scientific papers, their peer reviews, and the real author responses". **We do not create data from nothing. Instead, we use a "critique-and-refine" process**. In this process, the teacher model uses the author's "original response" ($r_{orig}$) as a blueprint or "high-quality reference". Then, it generates a new, better strategic response based on our TSR (ToM-Strategy-Response) framework.
> - We use a **mixture of powerful teacher models (such as GPT-4.1 and Claude 3.5) to create the data**. This helps to "reduce model-specific biases and add more style variety". This approach keeps our data close to real human interactions. At the same time, it adapts the data to the strategic level required by our TSR framework
> ### 3. **Performance-based pragmatic validation**
> For your main concern about "real-world effectiveness," we use the ”performance-based Validation". Our experimental results prove this approach works:
> - **Better than models trained on real data**: As we said before, our RebuttalAgent (5.42 score) does much better than the RebuttalFT baseline (6.35 score) trained on real data (Table 2 in our manuscript). This shows our refined data is better for training than the original raw data.
> - **Better than the teacher models**: Our agent does not simply copy the teachers; it performs better than them. In both automatic checks (Table 2) and the main human checks (Table 3), RebuttalAgent (9.42 auto, 9.57 human) scores much higher than strong models such as GPT-4.1 (8.50 auto, 8.99 human) and o3 (9.21 auto, 9.26 human).
> - **Strong performance on OOD datasets**: To answer the concern that our model only works in a simulation, we run OOD (out-of-domain) tests on our Rebuttal-test set. This dataset uses recent, real-world reviews from ICLR and NeurIPS that we collected by hand. It has no overlap with our training data. As Table 9 shows, RebuttalAgent (9.34 score) still gets the top score, beating all baselines on this pure, real-world OOD dataset.

---

> ### Author Response · Authors · 2025-11-19
> **(2/6) Responses by Authors**
>
> > **`Weakness 2`**: It is unclear whether GPT-4.1, o3, DeepSeek, and other baselines were evaluated under identical prompting and retrieval settings. It seems that only the Strategy-Prompt (GPT-4.1) baseline mimics the proposed strategic planning stage; other foundation models were not prompted to perform ToM inference. Thus, the reported gains may reflect differences in pipeline design or fine-tuning rather than inherent reasoning ability.
>
> `Response`: We sincerely thank you for the insightful feedback on baseline comparisons.  **We agree that evaluating strong models such as o3 and GPT-4 under our ToM framework is a highly constructive suggestion and have adopted your suggestion and conducted additional experiments**. We introduced two new powerful baselines to our comparison:
>
> - **TSR-o3**: This baseline uses the o3 model, guided by our complete ToM-Strategy-Response (TSR) framework as a prompt.
> - **TSR-GPT-4.1**: This baseline uses the GPT-4.1 model, also guided by our full TSR prompt.
>
> These new baselines, along with our other baselines **Self-Refined** (iterative self-reflection), **Strategy-Prompt** (GPT-4.1 mimicking our strategy), and **RebuttalFT** (trained on human data), create a more comprehensive and rigorous benchmark.
>
> **Table 2**. Automated Evaluation Results
> | Category | Rigor | | | Soundness | | | Significance | | | Presentation | | | Avg |
> |----------|-------|-|-|-----------|-|-|-------------|-|-|-------------|-|-|-----|
> | Metric      | C | P | Co | C | P | Co | C | P | Co | C | P | Co | |
> | o3       | 9.00 | 8.99 | 9.55 | 8.84 | 8.78 | **9.45** | 8.58 | 8.43 | 9.22 | 9.34 | 9.12 | 9.50 | 9.21 |
> |  GPT-4.1  | 8.34 | 7.86 | 8.80 | 8.27 | 7.79 | 8.62 | 8.05 | 7.28 | 8.20 | 8.91 | 8.57 | 9.42 | 8.50 |
> | DeepSeek-R1 | 8.47 | 7.90 | 8.90 | 8.46 | 8.03 | 8.75 | 8.29 | 7.71 | 8.60 | 9.03 | 8.70 | 9.54 | 8.64 |
> | Gemini-2.5 | 7.89 | 6.91 | 6.63 | 8.06 | 7.41 | 7.26 | 7.87 | 7.09 | 6.89 | 8.56 | 8.11 | 8.83 | 7.75 |
> |  Self-Refined  | 8.55 | 8.08 | 9.04 | 8.47 | 8.04 | 8.88 | 8.19 | 7.56 | 8.52 | 9.08 | 8.75 | 9.59 | 8.72 |
> |  Strategy-Prompt | 8.26 | 7.41 | 8.32 | 8.33 | 7.77 | 8.51 | 8.13 | 7.41 | 7.95 | 8.85 | 8.44 | 9.46 | 8.37 |
> | TSR-o3 | 8.89 | **9.10** | **9.68** | 8.95 | 8.91 | 9.28 | 8.69 | **8.56** | 9.45 | 9.18 | **9.35** | 9.45 | 9.34 |
> |  TSR-GPT4.1  | 8.47 | 7.63 | 8.53 | 8.12 | 7.94 | 8.85 | 7.90 | 7.51 | 8.45 | 9.07 | 8.42 | 9.16 | 8.76 |
> | RebuttalFT  | 6.91 | 6.07 | 6.80 | 6.58 | 5.72 | 6.24 | 6.52 | 5.50 | 5.94 | 6.55 | 5.79 | 6.63 | 6.35 |
> | RebuttalAgent | **9.23** | 8.91 | 9.59 | **9.18** | **8.95** | 9.37 | **9.09** | 8.54 | **9.65** | **9.43** | 9.20 | **9.50** | **9.42** |
>
> As shown in the table above, these new experiments provide two key insights:
> - The results show that o3 + TSR significantly outperforms the standard o3, and GPT-4.1 + TSR likewise outperforms the standard GPT-4.1. This confirms that **our ToM-based framework can effectively guide LLMs to generate more strategic responses**.
> - **Our RebuttalAgent (8B) still significantly outperforms both of these powerful new baselines (o3 + TSR and GPT-4.1 + TSR)**. The RebuttalBench dataset and two-stage training process (SFT + Self-Reward RL)  are necessary and effective for unlocking the deep strategic capabilities required for this complex academic rebuttal task.
>
> We have updated Table 2 with the complete results from these new baselines. We believe these additions greatly strengthen our experimental evaluation and more robustly validate our contributions.

---

> ### Author Response · Authors · 2025-11-19
> **(3/6) Responses by Authors**
>
> > **`Weakness 3`**: All model outputs are required to include the explicit [object Object][object Object][object Object] structure, and the reward function also checks format adherence. This raises the question of whether the model mainly learns to reproduce the expected format rather than to perform genuine strategic reasoning.
>
> `Response`: Thank you for this insightful question. We clarify that format adherence is merely a prerequisite for parsing rather than the final learning objective.
>
> ### 1. **The structured output externalizes the model's internal reasoning**.
> Our TSR framework (Sections 4.1-4.3) is designed to compel the model to first decide how to respond (the Strategy, S) based on its understanding of the reviewer's mental state (the Theory-of-Mind, T), before generating the final reply (the Response, R). The SFT phase (Section 5.2) teaches the model this structured reasoning chain from RebuttalBench. The RL phase (Section 5.3) then uses this externalized structure to directly optimize the quality of the model's thinking.
>
> ### 2. **Our reward function in the RL phase prioritizes content quality and diversity over simple format adherence**.
> As detailed in Section 5.3 and Appendix I, the reward is a weighted sum where content is heavily favored:
> - **Format Adherence** ($R_{format}$) accounts for **only 10% of the reward**.
> - **Reasoning & Response Quality** ($R_{think}$ + $R_{resp}$) accounts for **60% of the reward**. These signals explicitly evaluate the quality inside the tags, such as the accuracy of the analysis, the soundness of the strategy, and the persuasiveness of the final response.
> - **Response Diversity** ($R_{div}$) accounts for the remaining **30%**. This reward discourages generic, templated replies by comparing the output against a set of predefined negative examples, rewarding responses that are semantically distinct.
>
> In summary, **the $\<analysis\>$, $\<strategy\>$ and $\<response\>$tags are not the end goal. They function as an essential scaffold that  encourages the model to expose its internal strategic reasoning**. This allows our $R_{think}$ reward to directly access, evaluate, and optimize the quality of that reasoning.

---

> ### Author Response · Authors · 2025-11-19
> **(4/6) Responses by Authors**
>
> > **`Weakness 4`**: Since the reinforcement learning stage uses a self-reward mechanism derived from the same model, it is possible that the agent learns to optimize for its own scoring heuristics instead of true persuasiveness.
>
> `Response`: Thank you for this insightful question. We designed this mechanism to be scalable while avoiding the risk of "policy collapse" that is common with fixed, external reward models. We address this potential issue of optimizing for its own heuristics on three levels.
>
> ### 1. **Mechanism design to prevent policy collapse**
> Our self-reward mechanism includes a key component by design: **R_div (Response Diversity)**. As shown in Appendix I, this reward accounts for 30% of the total weight. **It explicitly penalizes homogeneous outputs** by comparing the generated response against a set of predefined negative (templated) examples. This forces the model to balance exploitation (the R_resp reward) with exploration (the R_div reward), which fundamentally prevents the model from converging on a single, self-optimizing heuristic.
>
> ### 2. **New experiments confirm overfitting in external RMs**
> For empirical comparison, we supplemented our work with new experiments. **We trained the agent using powerful external RMs (Rebuttal-RM and GPT-4.1) instead of our self-reward mechanism**. The results  above confirm that external RMs do cause the agent to "overfit" the evaluator. For instance, the agent trained with Rebuttal-RM achieves the highest automatic score from Rebuttal-RM. However, when these externally-trained models are judged by human evaluators, their scores are significantly lower than our original agent. **This shows that fixed, external RMs cause the agent to optimize for specific heuristics, not "true persuasiveness"**.
>
> **Table 3**. Automated Evaluation Results
> | Category | Rigor | | | Soundness | | | Significance | | | Presentation | | | Avg |
> |----------|-------|-|-|-----------|-|-|-------------|-|-|-------------|-|-|-----|
> | Metric      | C | P | Co | C | P | Co | C | P | Co | C | P | Co | |
> | w RebuttalRM-reward | 9.39 | 9.35 | 9.51 | 9.40 | 9.32 | 9.29 | 9.53 | 8.95 | 9.70 | 9.61 | 9.45 | 9.89 | 9.45 |
> | w GPT4.1-reward | 9.33 | 9.24 | 8.85 | 9.32 | 9.16 | 9.82 | 9.35 | 9.07 | 9.30 | 9.24 | 9.38 | 9.18 | 9.35 |
> | RebuttalAgent   | 9.23 | 8.91     | 9.59 | 9.18     | 8.95     | 9.37     | 9.09     | 8.54     | 9.65 | 9.43  | 9.20      | 9.50      | 9.42 |
>
> **Table 4**. Human Evaluation Results
> | Model          | Attitude | Clarity | Persuasiveness | Constructiveness | Avg   |
> |----------------|----------|---------|----------------|------------------|-------|
> | w RebuttalRM-reward | 9.16 | 8.90 | 8.84 | 9.07 | 8.96 |
> | w GPT4.1-reward  | 9.92 | 9.62 | 9.28 | 9.54 | 9.59 |
> | RebuttalAgent    | 9.86 | 9.38 | 9.34 | 9.68 | 9.57 |
>
>
> ### 3. **Human evaluation confirms the effectiveness of our self-reward mechanism**
> “True persuasiveness" must be judged by humans. As shown in our Section 7.4 (Human Evaluation), the RebuttalAgent trained with our self-reward mechanism achieves an average score of 9.57 in a blind review. This result provides strong evidence that our self-reward mechanism, successfully aligns with true human preferences and that its optimization direction generalizes to expert human standards.
>
> **Table 5**. Human Evaluation Results
> | Model          | Attitude | Clarity | Persuasiveness | Constructiveness | Avg   |
> |----------------|----------|---------|----------------|------------------|-------|
> | o3             | 9.3      | 9.28    | 9.04           | 9.42             | 9.26  |
> | GPT-4.1        | 9.32     | 8.8     | 8.7            | 9.14             | 8.99  |
> | DeepSeek-R1    | 9.24     | 9.08    | 8.86           | 9.16             | 9.08  |
> | Qwen3-8B       | 8.88     | 8.6     | 8.12           | 8.40              | 8.50   |
> | RebuttalFT     | 7.38     | 6.8     | 6.3            | 6.5              | 6.75  |
> | RebuttalAgent  | **9.86** | **9.38** | **9.34**       | **9.68**         | **9.57** |
>
>
> Thank you for your valuable feedback, we have added these new experiments and updated the relevant descriptions and clarifications in our revised manuscript.

---

> ### Author Response · Authors · 2025-11-19
> **(5/6) Responses by Authors**
>
> > **`Question 1`**: The work does not show that generated rebuttals can actually influence reviewer judgments or acceptance outcomes. The "persuasiveness score" used by Rebuttal-RM serves only as a proxy, without evidence that it correlates with real reviewer behavior. While a full-scale deployment would be difficult, would it be possible to conduct a small-scale testing to verify whether the generated rebuttals are genuinely persuasive to expert reviewers
>
> `Response`: Thank you for this insightful question. We fully agree with your observation that automated evaluations are essentially proxy indicators. **We are pleased to clarify that we have already performed the human evaluation you suggested in Section 7.4**, as shown in Table below.
>
> ### 1.  **The effectiveness of Rebuttal-RM as an evaluator**
> It is worth noting that we developed Rebuttal-RM to conduct reliable and efficient automated evaluation of rebuttal quality. We recognized that for it to be a meaningful tool, it must first align highly with human expert judgment. Therefore, **in Section 7.1, we specifically validated Rebuttal-RM's consistency with human preferences**. As shown in Table 1, Rebuttal-RM achieved the highest average score (0.812) for alignment with human judgments, significantly outperforming all baseline models, including GPT-4.1 (0.745). This demonstrates that **Rebuttal-RM is a validated and more reliable evaluation proxy than other large models**.
>
> ### 2. **Human expert evaluation**
> More importantly, **the human expert evaluation is detailed in Section 7.4**. It utilized 100 randomly sampled comments, balanced between in-domain and out-of-domain instances . Each response was evaluated blindly by three annotators, who possessed "at least three years of research experience in AI/ML and prior reviewing experience in top-tier conferences" . This process demonstrated high reliability, with a strong inter-annotator agreement (Cohen's $\kappa = 0.79$).
>
> **Table 6**. Human evaluation on different models
> | Model          | Attitude | Clarity | Persuasiveness | Constructiveness | Avg   |
> |----------------|----------|---------|----------------|------------------|-------|
> | o3             | 9.3      | 9.28    | 9.04           | 9.42             | 9.26  |
> | GPT-4.1        | 9.32     | 8.8     | 8.7            | 9.14             | 8.99  |
> | DeepSeek-R1    | 9.24     | 9.08    | 8.86           | 9.16             | 9.08  |
> | Qwen3-8B       | 8.88     | 8.6     | 8.12           | 8.4              | 8.5   |
> | RebuttalFT     | 7.38     | 6.8     | 6.3            | 6.5              | 6.75  |
> | RebuttalAgent  | **9.86** | **9.38** | **9.34**       | **9.68**         | **9.57** |
>
> As presented in table above (from our manuscript), the human evaluation results decisively confirm the clear superiority of RebuttalAgent. **Our model achieved the highest average score of 9.57, establishing a significant lead over the strongest baselines, including o3 and GPT-4.1**. This advantage is comprehensive, as RebuttalAgent outperformed all other models across all four evaluation dimensions. Specifically regarding your concern about "Persuasiveness," RebuttalAgent achieved a score of 9.34, representing a 7.36% improvement over the GPT-4.1 baseline. This finding, combined with high scores in other metrics, confirms that RebuttalAgent is by far the most effective and balanced model.
>
> In summary, while Rebuttal-RM is a proxy metric, we have experimentally shown (in Section 7.1) that it is a reliable proxy that aligns strongly with human expert judgment. More importantly, we have directly confirmed (in Section 7.4) through a blind, expert human evaluation—the very "small-scale test" you suggested—that RebuttalAgent is genuinely persuasive to domain experts. We will emphasize this point more clearly in the revised manuscript to address your important concern.

---

> ### Author Response · Authors · 2025-11-19
> **(6/6) Responses by Authors**
>
> > **`Question 2`**: The proposed approach is trained on Qwen3-8B and Llama-3.1-8B. Can the TSR pipeline and self-reward mechanism transfer effectively to other backbones, such as smaller models (<8B)?
>
> `Response`: Thank you for your valuable question. **We conducted two sets of experiments to validate the transfer effectiveness of our TSR pipeline and self-reward mechanism to other backbone models, including smaller ones**. We have summarized these results in the table below:
>
> **Table 7**. Automated Evaluation Results
> | Category | Rigor | | | Soundness | | | Significance | | | Presentation | | | Avg |
> |----------|-------|-|-|-----------|-|-|-------------|-|-|-------------|-|-|-----|
> | Metric      | C | P | Co | C | P | Co | C | P | Co | C | P | Co | |
> | Qwen3-4B   | 7.84 | 7.05 | 7.42 | 7.77 | 6.98 | 6.99 | 7.72 | 6.69 | 6.83 | 8.48 | 8.02 | 8.66 | 7.69 |
> | Qwen3-8B   | 7.96 | 7.33 | 8.18 | 7.84 | 7.11 | 7.76 | 7.68 | 6.73 | 7.39 | 8.51 | 8.08 | 8.87 | 7.96 |
> | Llama-3.1-8B   | 7.77 | 6.69 | 7.32 | 7.71 | 6.76 | 7.02 | 7.54 | 6.30 | 6.49 | 8.12 | 7.42 | 8.25 | 7.44 |
> | RebuttalAgent_llama | 9.23 | 9.10 | 9.16 | 9.29 | 9.11 | 9.24 | 9.16 | 8.67 | 9.05 | 9.57 | 9.35 | 9.39 | 9.20 |
> | RebuttalAgent_qwen4b  | 8.79 | 8.54 | 9.73 | 8.60 | 8.24 | 9.44 | 8.32 | 7.84 | 9.17 | 9.12 | 8.76 | 9.72 | 8.98 |
> | **RebuttalAgent_qwen8b** | 9.23 | 8.91 | 9.59 | 9.18 | 8.95 | 9.37 | 9.09 | 8.54 | 9.65 | 9.43 | 9.20 | 9.50 | **9.42** |
>
> The results show that:
> - **Transferability to Different Model Families** (8B Size): As shown in the ablation study in our paper (Table 2), we evaluated our full framework on the Llama-3.1-8B backbone. The results show that the standard Llama-3.1-8B baseline scored an average of 7.44. After applying our method (shown as w/ Llama-based), the average score significantly increased to 9.20. This confirms our method's effectiveness on different model families of a similar size.
> - **Transferability to Smaller Models** (<8B) (Newly Added): To further address your suggestion, we added new experiments on the smaller Qwen3-4B model. As shown in the new results table, the Qwen3-4B baseline model achieved an average score of 7.69. When our TSR pipeline and self-reward mechanism were applied (RebuttalAgent_qwen4b), the average score rose to 8.98.
>
> The data shows that our framework provides strong generalizability and significant performance improvements when transferred to both Llama-3.1-8B (a different 8B model, with a gain from 7.44 to 9.20) and Qwen3-4B (a smaller <8B model, with a gain from 7.69 to 8.98). **These results collectively demonstrate that the effectiveness of our TSR pipeline and self-reward mechanism is not limited to a specific backbone**. The framework serves as a general strategy that can be successfully transferred to other models, including smaller ones.
>
> We have added these new experimental results for the Qwen3-4B model and the corresponding analysis to the revised manuscript.

---

> > ### Comment · Reviewer_LugC · 2025-11-25
> >
> > Thank you for your response. I will maintain my current positive evaluation of this paper.

---

> > > ### Author Response · Authors · 2025-11-25
> > >
> > > We appreciate your time and your decision to maintain the positive evaluation. **We highly value your constructive suggestions. If there are any remaining concerns that we could address to further strengthen your assessment, we are happy to answer them**. We remain available for any further discussion.

---

> ### Author Response · Authors · 2025-11-28
>
> Dear Reviewer LugC,
>
> Thank you for your thoughtful and constructive feedback on our paper. We have carefully addressed your comments and suggestions, which have significantly helped us enhance the rigor and clarity of our work. We also appreciate the generally positive feedback regarding our work.
>
> **We have uploaded a revised version of the manuscript** where major changes are highlighted in blue. In the revised manuscript, we expanded our evaluation by updating Table 2 with new baselines and ablation studies to validate model robustness. We also clarified methodological details and moved key details into the main text for better readability.
>
> We believe these revisions fully address the questions raised. If there are any further questions or if additional clarification is needed, we would be more than happy to answer them.
>
> Best regards,
>
> Authors of Paper Submission 8600

---

### Official Review · Reviewer_d1nh · 2025-10-31

**Soundness:** 3
**Presentation:** 3
**Contribution:** 3
**Rating:** 6
**Confidence:** 3

**Summary:**

This work focuses on the academic rebuttal task — how authors can comprehensively address reviewers’ comments, resolve their concerns, and persuade them through well-reasoned responses. This is an interesting and meaningful direction, representing an important step for LLMs toward academic reasoning. The paper introduces a ToM-based reasoning framework, a rebuttal benchmark for instruction tuning, a RebuttalJudge model for response evaluation, and further integrates reinforcement learning (RL) to optimize the RebuttalAgent. However, from a technical perspective, the paper does not present novel or compelling innovations; rather, it seems to combine existing techniques to tackle the defined task.

**Strengths:**

1. The task itself is very interesting, the motivation is clear, and the paper is well written.

2. The study provides several valuable resources, including the RebuttalBench benchmark and the RebuttalJudge evaluation model.

3. The paper offers a complete framework for developing a RebuttalAgent and presents comprehensive experiments that demonstrate its effectiveness.

**Weaknesses:**

1. The main concern is the lack of technical innovation.  I am curious whether using RebuttalJudge as a reward function to guide RL training would lead to improved performance under human evaluation. This would be an insightful experiment, showing that both the benchmark and the judge model can effectively guide the optimization of the RebuttalAgent.

2. Another concern lies in the baseline comparisons, which appear somewhat weak. While I am not deeply familiar with prior work specifically on academic rebuttal, the baselines used here seem to rely on relatively simple or earlier reasoning frameworks. Moreover, since this work involves a significant training cost, it would be helpful to include stronger baselines such as o3 or GPT-4 evaluated under the proposed ToM framework — these would serve as powerful non-trained baselines for comparison.

3. The ablation studies could be more detailed and targeted. They should aim to answer key questions — for example, how well does a model perform when trained solely on RebuttalBench? It would also be beneficial to conduct ablations separately under the SFT and RL settings rather than only presenting overall results.

**Questions:**

Refer to our proposed weakness.

---

> ### Author Response · Authors · 2025-11-19
> **(1/3) Responses by Authors**
>
> **Thanks for your careful and insightful reviews.**
>
> > **`Weakness 1`**: The main concern is the lack of technical innovation. I am curious whether using RebuttalJudge as a reward function to guide RL training would lead to improved performance under human evaluation. This would be an insightful experiment, showing that both the benchmark and the judge model can effectively guide the optimization of the RebuttalAgent.
>
> `Response`: We sincerely thank you for this insightful comment. **Following this suggestion, we conducted a new ablation study and trained two new model variants** and compared them against our original RebuttalAgent:
> - w RebuttalRM as reward model: An agent trained using the Rebuttal-RM evaluator as the direct reward function.
> - w GPT4.1 as reward model: An agent trained using GPT-4.1 as the reward function, serving as a powerful external baseline.
>
> **Table 1**. Automated Evaluation Results
> | Category | Rigor | | | Soundness | | | Significance | | | Presentation | | | Avg |
> |----------|-------|-|-|-----------|-|-|-------------|-|-|-------------|-|-|-----|
> | Metric      | C | P | Co | C | P | Co | C | P | Co | C | P | Co | |
> | w RebuttalRM-reward | 9.39 | 9.35 | 9.51 | 9.40 | 9.32 | 9.29 | 9.53 | 8.95 | 9.70 | 9.61 | 9.45 | 9.89 | 9.45 |
> | w GPT4.1-reward | 9.33 | 9.24 | 8.85 | 9.32 | 9.16 | 9.82 | 9.35 | 9.07 | 9.30 | 9.24 | 9.38 | 9.18 | 9.35 |
> | RebuttalAgent   | 9.23 | 8.91     | 9.59 | 9.18     | 8.95     | 9.37     | 9.09     | 8.54     | 9.65 | 9.43  | 9.20      | 9.50      | 9.42 |
>
> **Table 2**. Human Evaluation Results
> | Model          | Attitude | Clarity | Persuasiveness | Constructiveness | Avg   |
> |----------------|----------|---------|----------------|------------------|-------|
> | w RebuttalRM-reward | 9.16 | 8.90 | 8.84 | 9.07 | 8.96 |
> | w GPT4.1-reward  | 9.92 | 9.62 | 9.28 | 9.54 | 9.59 |
> | RebuttalAgent    | 9.86 | 9.38 | 9.34 | 9.68 | 9.57 |
>
> The **w RebuttalRM as reward model achieved the highest score when evaluated by Rebuttal-RM**. This is expected, as it was "teaching to the test”, directly optimizing for the evaluator's criteria. **However, its poor performance in the human evaluation highlights the risk of this overfitting**. As shown in the table, the agent trained with w RebuttalRM-reward (8.96 Avg) performed significantly worse than our original RebuttalAgent (9.57 Avg) under human evaluation. This suggests that while Rebuttal-RM is a strong evaluator, using it as a direct reward signal causes the agent to overfit to its specific scoring biases, which do not perfectly generalize to nuanced human preferences.
>
> Conversely, **our original RebuttalAgent, using its self-reward mechanism, achieved performance nearly identical to the agent trained with GPT-4.1 as a reward**. This finding strongly validates our proposed self-reward approach, demonstrating that it is robust and achieves results comparable to a powerful, external proprietary model, thereby supporting its technical contribution.
>
> Thank you once again for this perceptive suggestion. This experiment has provided a valuable ablation study that strengthens the justification for our self-reward mechanism. We have added these results and findings to our manuscript.

---

> ### Author Response · Authors · 2025-11-19
> **(2/3) Responses by Authors**
>
> > **`Weakness 2`**: Another concern lies in the baseline comparisons, which appear somewhat weak. While I am not deeply familiar with prior work specifically on academic rebuttal, the baselines used here seem to rely on relatively simple or earlier reasoning frameworks. Moreover, since this work involves a significant training cost, it would be helpful to include stronger baselines such as o3 or GPT-4 evaluated under the proposed ToM framework — these would serve as powerful non-trained baselines for comparison.
>
> `Response`: We sincerely thank you for the insightful feedback on baseline comparisons.  **We agree that evaluating strong models such as o3 and GPT-4 under our ToM framework is a highly constructive suggestion and have adopted your suggestion and conducted additional experiments**. We introduced two new powerful baselines to our comparison:
>
> - **TSR-o3**: This baseline uses the o3 model, guided by our complete ToM-Strategy-Response (TSR) framework as a prompt.
> - **TSR-GPT-4.1**: This baseline uses the GPT-4.1 model, also guided by our full TSR prompt.
>
> These new baselines, along with our other baselines **Self-Refined** (iterative self-reflection), **Strategy-Prompt** (GPT-4.1 mimicking our strategy), and **RebuttalFT** (trained on human data), create a more comprehensive and rigorous benchmark.
>
> **Table 3**. Automated Evaluation Results
> | Category | Rigor | | | Soundness | | | Significance | | | Presentation | | | Avg |
> |----------|-------|-|-|-----------|-|-|-------------|-|-|-------------|-|-|-----|
> | Metric      | C | P | Co | C | P | Co | C | P | Co | C | P | Co | |
> | o3       | 9.00 | 8.99 | 9.55 | 8.84 | 8.78 | **9.45** | 8.58 | 8.43 | 9.22 | 9.34 | 9.12 | 9.50 | 9.21 |
> |  GPT-4.1  | 8.34 | 7.86 | 8.80 | 8.27 | 7.79 | 8.62 | 8.05 | 7.28 | 8.20 | 8.91 | 8.57 | 9.42 | 8.50 |
> | DeepSeek-R1 | 8.47 | 7.90 | 8.90 | 8.46 | 8.03 | 8.75 | 8.29 | 7.71 | 8.60 | 9.03 | 8.70 | 9.54 | 8.64 |
> | Gemini-2.5 | 7.89 | 6.91 | 6.63 | 8.06 | 7.41 | 7.26 | 7.87 | 7.09 | 6.89 | 8.56 | 8.11 | 8.83 | 7.75 |
> |  Self-Refined  | 8.55 | 8.08 | 9.04 | 8.47 | 8.04 | 8.88 | 8.19 | 7.56 | 8.52 | 9.08 | 8.75 | 9.59 | 8.72 |
> |  Strategy-Prompt | 8.26 | 7.41 | 8.32 | 8.33 | 7.77 | 8.51 | 8.13 | 7.41 | 7.95 | 8.85 | 8.44 | 9.46 | 8.37 |
> | TSR-o3 | 8.89 | **9.10** | **9.68** | 8.95 | 8.91 | 9.28 | 8.69 | **8.56** | 9.45 | 9.18 | **9.35** | 9.45 | 9.34 |
> |  TSR-GPT4.1  | 8.47 | 8.23 | 8.54 | 8.12 | 7.94 | 8.85 | 7.90 | 7.51 | 8.45 | 9.07 | 8.42 | 9.16 | 8.86 |
> | RebuttalFT  | 6.91 | 6.07 | 6.80 | 6.58 | 5.72 | 6.24 | 6.52 | 5.50 | 5.94 | 6.55 | 5.79 | 6.63 | 6.35 |
> | RebuttalAgent | **9.23** | 8.91 | 9.59 | **9.18** | **8.95** | 9.37 | **9.09** | 8.54 | **9.65** | **9.43** | 9.20 | **9.50** | **9.42** |
>
> As shown in the table above, these new experiments provide two key insights:
> - The results show that o3 + TSR significantly outperforms the standard o3, and GPT-4.1 + TSR likewise outperforms the standard GPT-4.1. This confirms that **our ToM-based framework can effectively guide LLMs to generate more strategic responses**.
> - **Our RebuttalAgent (8B) still significantly outperforms both of these powerful new baselines (o3 + TSR and GPT-4.1 + TSR)**. The RebuttalBench dataset and two-stage training process (SFT + Self-Reward RL)  are necessary and effective for unlocking the deep strategic capabilities required for this complex academic rebuttal task.
>
> We have updated Table 2 with the complete results from these new baselines. We believe these additions greatly strengthen our experimental evaluation and more robustly validate our contributions.

---

> ### Author Response · Authors · 2025-11-19
> **(3/3) Responses by Authors**
>
> > **`Weakness 3`**: The ablation studies could be more detailed and targeted. They should aim to answer key questions — for example, how well does a model perform when trained solely on RebuttalBench? It would also be beneficial to conduct ablations separately under the SFT and RL settings rather than only presenting overall results.
>
> `Response`: Thank you for your valuable advice regarding our ablation studies. We apologize for the lack of clarity in our initial presentation. In fact, **we had already conducted these detailed and targeted ablation experiments in the original Table 2**. To improve clarity and make the section more logical, we have reorganized our ablation studies as follows.
>
> - **SFT-only (w/o RL)**: This corresponds to the w/o RL baseline in the 'Training Ablation' section of the original Table 2. As you suggested, this model is "trained solely on RebuttalBench" and evaluates the SFT model without the RL stage.
> - **RL-only (w/o SFT)**: This corresponds to the w/o SFT baseline, which assesses the effect of performing RL without the foundational SFT stage.
> - **w/ Llama-based**: This experiment uses Llama-3.1-8B as the base model to test our framework's generalizability.
> - **w/ DPO**: This replaces our GRPO algorithm with DPO as the optimization algorithm for the RL stage.
> - **w/o $R_{Analysis}$, w/o $R_{Response}$, w/o $R_{Format}$**: These are the ablation experiments for the individual reward components of our self-reward mechanism during the RL stage.
>
> **Table 4**. Automated Evaluation Results
> | Category | Rigor | | | Soundness | | | Significance | | | Presentation | | | Avg |
> |----------|-------|-|-|-----------|-|-|-------------|-|-|-------------|-|-|-----|
> | Metric      | C | P | Co | C | P | Co | C | P | Co | C | P | Co | |
> | w Llama-based | 9.23 | 9.10 | 9.16 | 9.29 | 9.11 | 9.24 | 9.16 | 8.67 | 9.05 | 9.57 | **9.35** | 9.39 | 9.20 |
> | w DPO | 8.47 | 8.13 | 9.36 | 8.32 | 7.92 | 9.00 | 8.11 | 7.57 | 8.82 | 8.94 | 8.55 | 9.46 | 8.68 |
> | SFT-only (w/o RL) | 8.20 | 7.60 | 8.42 | 8.17 | 7.60 | 8.28 | 8.02 | 7.31 | 7.84 | 8.76 | 8.34 | 9.16 | 8.27 |
> | RL-only (w/o SFT) | 8.63 | 8.27 | 9.42 | 8.47 | 8.07 | 9.01 | 8.21 | 7.56 | 8.34 | 9.05 | 8.71 | 9.61 | 8.79 |
> | w/o $R_{Analysis}$ | 9.25 | 9.23 | 9.79 | 9.20 | 9.18 | 9.39 | 9.00 | 8.87 | 9.27 | 9.59 | 9.41 | 9.45 | 9.23 |
> | w/o $R_{Response}$ | 8.51 | 7.90 | 9.02 | 8.41 | 7.91 | 8.63 | 8.17 | 7.51 | 8.25 | 9.05 | 8.68 | 9.61 | 8.63 |
> | w/o $R_{Format}$ | 9.06 | 8.91 | 9.22 | 9.04 | 8.74 | 9.30 | 8.88 | 8.29 | 9.67 | 9.37 | 9.14 | 9.35 | 9.32 |
> | w $R_{Dist. weights}$ | 9.08 | 8.54 | 9.53 | 9.04 | 8.63 | 9.23 | 9.05 | 8.32 | 9.85 | 9.34 | 9.08 | 9.38 | 9.27 |
>
> As shown in the table above, performance drops significantly when omitting core training stages. Specifically, the SFT-only (w/o RL) model performs substantially worse than the full RebuttalAgent, demonstrating the critical importance of our self-reward RL phase for strategic optimization. Ablating the RL reward signals confirms their importance. Among all signals, the reward for the final response quality proved to be the most impactful, as its removal caused the average score to drop to 8.63. **We have updated the manuscript to reflect this clearer, reorganized structure for the ablation studies**.

---

> ### Author Response · Authors · 2025-11-26
>
> Dear Reviewer d1nh,
>
> Thank you once again for your thoughtful feedback on our paper. We appreciate the time and effort you have invested in reviewing our work.
>
> As we are nearing the end of the rebuttal period, we wanted to kindly follow up to ensure that you’ve had a chance to review our response to your comments. We hope that our rebuttal has addressed your concerns satisfactorily. If there are any additional points or further clarifications required, we would be happy to provide them promptly.
>
> Additionally, if our response has sufficiently addressed your concerns, we would greatly appreciate if you could consider reflecting this in your evaluation, including revisiting your score if appropriate.
>
> Thank you for your time and understanding. We greatly value your feedback and look forward to hearing from you.
>
> Best regards,
>
> Authors of Paper Submission 8600

---

> ### Author Response · Authors · 2025-11-28
>
> Dear Reviewer d1nh,
>
> Thank you for your thoughtful and constructive feedback on our paper. We have carefully addressed your comments and suggestions, which have significantly helped us enhance the rigor and clarity of our work. We also appreciate the generally positive feedback regarding our work.
>
> **We have uploaded a revised version of the manuscript** where major changes are highlighted in blue. In the revised manuscript, we expanded our evaluation by updating Table 2 with new baselines and ablation studies to validate model robustness. We also clarified methodological details and moved key details into the main text for better readability.
>
> We believe these revisions fully address the questions raised. If there are any further questions or if additional clarification is needed, we would be more than happy to answer them.
>
> Best regards,
>
> Authors of Paper Submission 8600

---

### Official Review · Reviewer_snQL · 2025-10-31

**Soundness:** 2
**Presentation:** 2
**Contribution:** 3
**Rating:** 4
**Confidence:** 3

**Summary:**

The paper is on using Theory of Mind (ToM) to help with effective persuasion in the academic rebuttal scenario. Such goal is to be achieved through three stages. First, motivated by ToM, the paper proposes a 'RebuttalAgent' framework using a process of reviewer perspective inference, strategy and persuasive responses (TSR). Then the 'RebuttalBench' is formed as the dataset for training, with which both  supervised fine-tuning and reinforcement learning are utilized. Finally, a 'Rebuttal-RM' is also trained as the model to measure the responses generated by the above agent. The advantage of such agent is shown, compared to baseline models, across several dimensions of the responses.

**Strengths:**

> **Originality**
- The framework is motivated by ToM.

> **Significance**
- The improvement, compared to the list of selected baseline models, is significant.

**Weaknesses:**

> **Originality**
- A discussion on the difference in methodology, compared to those mentioned as Agent-based Methods, could be helpful.

> **Quality**
- Section 3.1 Comment Extraction: As the authors suggest, the raw review comments can be unstructured with varying noises, styles or formats, then the proposed LLM-as-Extractor may need some systematic verification on its performance.
- Section 3.1 Comment Extraction (continued): According to the prompt in Figure 6, there could be several concerns. First, the justification of the categories listed in Macro/Micro-Analysis. Second, why never split a comment item? What if multiple directions are covered? Overall the concern is the fixed framework of dimensions and criteria that may not always guarantee a satisfying comprehension.
- Such comprehension issue can also exist in Section 4.1 Reviewer Profile Modeling.
- According to the reviewer's understanding (feel free to correct), the response is generated comment-wise. Then how to guarantee the consistency and avoid redundancy when integrating all responses together?
- Whether it is justified to use SFT-tuned model in its own evaluation/training in Section 5.3, especially how to make sure the evaluation aligns with the expected direction of improvement?


> **Clarity**
- A more detailed review of the Theory-of-Mind is necessary before introducing the proposed methods.
- It is a bit unclear where the original response, $r_\textrm{orig}$, is from. If it is given as part of the dataset, then in the final implementation of trained model, will this part be excluded?
- Line 316-323: The paragraph contains two copies of the same sentences.
- Section 7.1: The notations of metrics used should be explained in the main paper.

**Questions:**

- Could the authors elaborate how ToM is explicitly used? According to the reviewer's understanding, such details are rather limited in the main paper.
- When evaluating the responses in the main experiment, is the score an average of comment-wise responses or considers the quality of the concatenated responses against each complete review?

---

> ### Author Response · Authors · 2025-11-19
> **(1/7) Responses by Authors**
>
> **Thanks for your careful and insightful reviews.**
> > **`Weakness 1`**: A discussion on the difference in methodology, compared to those mentioned as Agent-based Methods, could be helpful.
>
> `Response`: Thank you for your valuable feedback. **We have introduced the different baselines in Section 7.2. Due to space constraints in the original manuscript, the discussion was concise. Below we provide a more detailed elaboration on the differences between our method and these baselines**.
>
> Our agent-based baselines, RebuttalFT, Self-Refined, and Strategy-Prompt, represent different approaches ranging from direct imitation to iterative refinement. Below are the detailed methodology descriptions for the original baselines:
>
> - **RebuttalFT** (Direct Fine-tuning): RebuttalFT represents "naive imitation learning". It is a model directly supervised fine-tuned on the $R^{2}$-rebuttal dataset2.
> - **Self-Refined**: This is a general, iterative "generate-reflect" refinement method. It generates an initial response and then iteratively refines it via self-reflection.
> - **Strategy-Prompt**: This method "mimics our methodology by first generating a strategic plan... before writing". It uses GPT-4.1 as its backbone.
> To provide an even stronger point of comparison, we have also added new experiments:
> - **TSR**: using TSR-GPT4.1, which applies our full TSR (ToM-Strategy-Response) framework to the powerful GPT-4.1 backbone.
>
>  To provide a clearer comparison of the different approaches, we have summarized their key methodological aspects and performance in the table below.
>
>  | Method | Backbone Model | Core Methodology | Reviewer Modeling (ToM) | Training Paradigm | Auto. Avg Score (R2-test) | Human Eval Avg Score | OOD Avg Score (Rebuttal-test) |
> |--------|------------------|-------------------------|-------------------|----------------|--------------------------|---------------------|------------------------------|
> | Strategy-Prompt | GPT-4.1 | Strategy-then-Response | No (Implicit) | Prompting | 8.37 | 8.42 | 8.50 |
> | Self-Refined | GPT-4.1 | Generate-Reflect-Iterate | No | Prompting | 8.72 | 8.64 | 8.99 |
> | TSR-GPT4.1 (New) | GPT-4.1 | TSR (ToM-Strategy-Response) | Yes (Explicit ToM Modeling) | Prompting | 8.76 | 9.15 | N/A |
> | RebuttalFT | Qwen3-8B | End-to-End SFT Imitation | No | SFT | 6.35 | 6.75 | 6.23 |
> | RebuttalAgent | Qwen3-8B | TSR (ToM-Strategy-Response) | Yes (Explicit ToM Modeling) | SFT (TSR chain) + RL (Self-Reward) | 9.42 | 9.57 | 9.34 |
>
> **RebuttalAgent is the only framework that explicitly integrates a Theory of Mind (ToM) component to model the reviewer's mental state**. Second, it utilizes a unique two-stage SFT+RL training paradigm, which moves beyond the simple prompting or direct SFT used by the baselines. The performance results confirm that our 8B RebuttalAgent significantly outperforms all baselines.
>
> We have added this detailed discussion, the comparison table, and the full results for the new TSR-GPT4.1 experiments in our revised manuscript. Thank you again for your constructive feedback.

---

> ### Author Response · Authors · 2025-11-19
> **(2/7) Responses by Authors**
>
> > **`Weakness 2`**: Section 3.1 Comment Extraction: As the authors suggest, the raw review comments can be unstructured with varying noises, styles or formats, then the proposed LLM-as-Extractor may need some systematic verification on its performance.
>
> `Response`: We completely agree that verifying the LLM-as-Extractor is essential, and actually we have conducted this verification. **As we noted at the end of Section 3.1, our extractor "achieves a 98% accuracy in comment extraction", with the full details of this validation process provided in Appendix B**.
>
> We designed a specific prompt that instructs an LLM-as-Extractor to parse the full raw review text. This prompt tasks the model to "identify and separate each distinct point of criticism", effectively segmenting a single, unstructured review into a list of discrete, actionable comments. We applied this method to parse all the reviews from our source dataset, which resulted in over 200,000 distinct comment-response pairs.
>
> To thoroughly validate the extractor's reliability, we randomly sampled 100 raw reviews. **This manual verification was conducted by three PhD students, all of whom have experience in peer review and authoring rebuttals**. They were tasked with comparing the full, original review text against the list of comments generated by our extractor. The goal was to ensure each extracted comment accurately and completely captured a distinct, actionable point of criticism. The annotators achieved a high inter-annotator agreement, with a Cohen's kappa score of 0.89.
>
> **Our analysis of this process showed that over 98 percent of the extracted comments were both complete and well-aligned with the reviewers' intended points**, with only 2 percent containing minor segmentation errors or incorporating redundant information. This high accuracy demonstrates that our LLM-as-Extractor framework is robust in handling diverse writing styles and unstructured formats.
>
> We thank you for this feedback. In our revised manuscript, **we will move these key details about our validation methodology** (including the annotator qualifications and agreement score) from Appendix B into Section 3.1 to make the reliability of our data preparation pipeline clearer to the reader.

---

> ### Author Response · Authors · 2025-11-19
> **(3/7) Responses by Authors**
>
> > **`Weakness 3 and 4`**: Section 3.1 Comment Extraction (continued). Such comprehension issue can also exist in Section 4.1 Reviewer Profile Modeling.
>
> `Response`: We sincerely thank you for your insightful comments. We have introduced the justification for the Macro/Micro-Analysis categories in Section 4.1 and provided detailed specifications in Table 4 of the Appendix.
> ### 1. **Rationale for the Macro/Micro-Analysis Categories**
> - **For Macro-level**: This level focuses on your overall decision-making state. We referred to the official reviewer guidelines from top-tier conferences such as ICLR and NeurIPS. These guidelines explicitly require reviewers to form a holistic judgment, such as providing a final recommendation (which maps to our "Overall Stance") and assessing their own expertise level (which maps to "Reviewer Expertise"). By modeling these dimensions, our RebuttalAgent can infer your overall stance and formulate a globally optimal persuasion strategy.
> - **For Micro-level**: This level focuses on the technical intent behind each specific comment. These categories are also derived from the core evaluation criteria within those same reviewer guidelines. Reviewers are explicitly asked to assess a paper's quality along several key dimensions, which our micro-categories codify : Significance, Methodology, Experimental Rigor, and Presentation.
> ### 2. **What if multiple directions are covered?**
> The core objective of our Comment Extraction module (Section 3.1) is to identify and separate each distinct point of criticism to segment a review into discrete, actionable comments. This is precisely intended to address the multiple directions issue you raised. We conducted a manual sampling and statistical analysis of 100 raw reviews and found that **87%** of the distinct points of feedback already contained only one core concern. Based on this finding, the primary task of our LLM-as-Extractor (Section 3.1) is to accurately identify and separate these pre-existing, relatively independent concerns, extracting them from noise such as salutations and summaries.
>
> ### 3. **Effectiveness of our dimensions and criteria**
> We have conducted empirical evaluations to verify the effectiveness of our ToM module.
> - **Human Evaluation**: In our human evaluation (Table 3), RebuttalAgent achieved the highest average score (95.57), significantly surpassing powerful and flexible baselines such as o3. To further address your concerns, we randomly sampled 100 comments and the review profile analyses generated by RebuttalAgent, and had them manually evaluated by a panel of three PhDs with peer-review experience. The results showed that **93%** of the review profiles were judged to be reasonable and accurate.
>
> - **Ablation Studies**: We directly quantified the contribution of each component in our ablation studies (Table 2, Data Ablation rows). When we removed the Theory of Mind analysis module (w/o ToM), the model's average score dropped from 9.42 to 9.04. When we removed the Strategy generation module (w/o Strategy), the score also dropped from 9.42 to 9.31. This proves that our TSR pipeline is an active and necessary reasoning scaffold that significantly enhances response quality.
> - **Generalization Experiments**: The results show that equipping models like Qwen3 and Llama3.1 with our T&S modules consistently improved their performance across all evaluation categories. For example, adding the full T&S framework to Qwen3-8B boosted its "Presentation" score by 21.0%.
>
> > **`Weakness 5`**: How to guarantee the consistency and avoid redundancy when integrating all responses together?
>
> `Response`: Thank you for this insightful question. While our RebuttalAgent generates responses comment-wise to ensure precise handling of specific criticisms, we have designed the ToM framework to inherently address consistency:
> - **Global Consistency via Macro-Profiling**: As detailed in Section 4.1, our agent does not view comments in isolation. It first performs a Macro-level analysis of the entire review to construct a holistic mental model of the reviewer. This global profile acts as a shared condition for all individual responses, ensuring that the tone, argumentation style, and strategic concession/defense trade-offs remain consistent across the entire rebuttal.
> - **Strategic Alignment**: The Strategy generation phase (Section 4.2) formulates a plan based on this shared profile before generating text. This ensures that even if comments are processed sequentially, they are aligned with the same high-level persuasion strategy.
>
> We chose the comment-wise approach to align with modern peer-review platforms (e.g., OpenReview), which often structure rebuttals as threaded replies to specific points. This hierarchical design allows RebuttalAgent to deliver targeted responses to specific concerns while maintaining a cohesive and consistent strategic voice across the entire rebuttal. We will add this clarification in the next version of our paper.

---

> ### Author Response · Authors · 2025-11-19
> **(4/7) Responses by Authors**
>
> > **`Weakness 6`**: Whether it is justified to use SFT-tuned model in its own evaluation/training in Section 5.3, especially how to make sure the evaluation aligns with the expected direction of improvement?
>
> `Response`: Thank you for raising this critical question. We justify using the SFT-tuned model for its own evaluation based on two theoretical premises and specific methodological safeguards, which are then confirmed by our empirical results.
>
> ### 1. **Verification vs. Generation Gap**
> The rationale for using the SFT model as a judge lies in the well-observed phenomenon in LLMs that verification capability often exceeds generation capability [1,2,3]. (1) **Latent Knowledge**: During the SFT phase (Section 5.2), the model has been exposed to high-quality, expert-annotated samples from RebuttalBench. (2) **Domain Expertise**: An external generic reward model might lack the specific context of academic rebuttals (e.g., the nuance of "polite pushback"). Our $\mathcal{G}_{SFT}$ is specialized in this domain, making it a more sensitive critic than a generic model.
>
> ### 2. **Methodological Safeguards**
> To specifically address your concern about "aligning with the expected direction", we do not rely solely on the model's subjective scoring. We introduce external constraints:
> (1) **$R_{div}$ (Diversity Reward)**: This is the key to preventing mode collapse. By explicitly penalizing responses that are semantically similar to a set of negative/generic templates, we force the model to explore new and distinct solution spaces. (2) **$R_{format}$ (Hard Constraints)**: We enforce strict structural rules that act as an objective "guardrail," ensuring the model does not drift away from the required reasoning format.
>
> ### 3. **Empirical Validation**
> Our experiments confirm that these mechanisms successfully guide the model in the right direction. Crucially, we conducted a controlled comparison (see table below) where we replaced our self-reward mechanism with external reward models (Rebuttal-RM and GPT-4.1). The results show that RebuttalAgent (Self-Reward) achieves performance comparable to or better than agents trained with external rewards, validating that the SFT model is a competent and aligned judge for this specific task.
>
> **Table 1**. Comparing with External RMs
> | Category | Rigor | | | Soundness | | | Significance | | | Presentation | | | Avg |
> |----------|-------|-|-|-----------|-|-|-------------|-|-|-------------|-|-|-----|
> | Metric      | C | P | Co | C | P | Co | C | P | Co | C | P | Co | |
> | w RebuttalRM-reward | 9.39 | 9.35 | 9.51 | 9.40 | 9.32 | 9.29 | 9.53 | 8.95 | 9.70 | 9.61 | 9.45 | 9.89 | 9.45 |
> | w GPT4.1-reward | 9.33 | 9.24 | 8.85 | 9.32 | 9.16 | 9.82 | 9.35 | 9.07 | 9.30 | 9.24 | 9.38 | 9.18 | 9.35 |
> | RebuttalAgent   | 9.23 | 8.91     | 9.59 | 9.18     | 8.95     | 9.37     | 9.09     | 8.54     | 9.65 | 9.43  | 9.20      | 9.50      | 9.42 |
>
> **Comparing with External RMs**: We conducted new experiments comparing it against two external reward models: 1) our own Rebuttal-RM and 2) GPT-4.1.The results suggest that our SFT model, when guided by our composite reward function, is a more effective domain-expert judge for this task than even powerful external models.
>
> **Table 2**. Human Evaluation
> | Model          | Attitude | Clarity | Persuasiveness | Constructiveness | Avg   |
> |----------------|----------|---------|----------------|------------------|-------|
> | o3             | 9.3      | 9.28    | 9.04           | 9.42             | 9.26  |
> | GPT-4.1        | 9.32     | 8.8     | 8.7            | 9.14             | 8.99  |
> | RebuttalAgent  | **9.86** | **9.38** | **9.34**       | **9.68**         | **9.57** |
>
> **Human Evaluation (Table 3)**: This is our most important validation. In a blind test by human experts, our final RebuttalAgent achieved the highest average score. It outperformed all other models, including o3 and GPT-4.1, confirming that our training process successfully aligns the model with the expected direction of improvement as judged by human standards.
>
> Thank you for your feedback, We will add this clarification in the next version of our paper.
>
> [1] Judging llm-as-a-judge with mt-bench and chatbot arena, NeurIPS 2023
>
> [2] Reflexion: Language Agents with Verbal Reinforcement Learning, NeurIPS 2023
>
> [3] A theoretical understanding of self-correction through in-context alignment, NeurIPS 2024

---

> ### Author Response · Authors · 2025-11-19
> **(5/7) Responses by Authors**
>
> > **`Weakness 7`**:  A more detailed review of the Theory-of-Mind is necessary before introducing the proposed methods.
>
> `Response`: Thank you very much for your valuable feedback. While we briefly touched upon this concept in the Introduction, we recognize the need for a more comprehensive discussion. **In the revised manuscript, we have added a more thorough explanation of the core concept of ToM in Section 4.1** (before introducing our specific methods).
>
> Theory of Mind (ToM) is a core concept in cognitive science, referring to the ability to understand and reason about the differing beliefs, intentions, desires, and perspectives of others. Applying this concept to artificial intelligence has led to Machine Theory of Mind (MToM), which is an AI system's capacity to infer and model the mental states of human or AI teammates to support collaboration. Large language models such as GPT-4 have demonstrated stronger ToM-like reasoning capabilities. In our work, we extend MToM to the specific domain of academic rebuttal. Given the game-theoretic and information-asymmetric nature of the rebuttal process, modeling the reviewer's beliefs, knowledge background, and core concerns is particularly critical. Therefore, our proposed RebuttalAgent framework explicitly implements ToM through a Theory-of-Mind-Strategy-Response (TSR) pipeline, which first constructs a hierarchical reviewer profile to guide the subsequent formulation of strategy and response.
>
> Thank you again for your constructive suggestions.
>
>
>
>
> > **`Weakness 8`**:   It is a bit unclear where the original response, r_orig, is from. If it is given as part of the dataset, then in the final implementation of trained model, will this part be excluded?
>
> `Response`: We apologize for this ambiguity. **The `r_orig` (or "original response") is sourced from the $Re^{2}$-rebuttal dataset. This dataset contains "authentic author responses" written by the actual paper authors**.
>
> As you correctly inferred, `r_orig` **is used only during the data synthesis phase**, not during the final model's inference phase.
>
> In our paper, Section 4.3 details the process of how we created our training data (RebuttalBench). In this synthesis process, we used `r_orig` as a "contextual input". Its purpose was twofold:
>
> 1. It served as a high-fidelity source of context, similar to the retrieved paper chunks[.
> 2. It provided a high-quality reference for phrasing and structure, which the model used as a "blueprint" to refine and build upon for the final output. This allowed us to generate the improved responses (containing the full mind-strategy-response chain) for our training set.
>
> In the implementation of the final trained model (RebuttalAgent), `r_orig` is completely excluded. In short, `r_orig` was one of the "raw materials" used to *create* our high-quality training data, not an input used by the final RebuttalAgent during runtime.
>
> Thank you for pointing this out. We will add a clear clarification in the revised version of Section 4.3 to state that `r_orig` is used only for data synthesis and is not available during model inference.
>
>
> > **`Weakness 9`**: Line 316-323: The paragraph contains two copies of the same sentences.
>
> `Response`: Thank you for your careful reading and for catching this error. We apologize for this oversight, which was a copy-paste mistake made during drafting. **We have now removed the duplicated sentences from the paragraph in the revised manuscript. We have also conducted a  thorough proofread of the entire manuscript** to identify and correct any remaining typographical errors to ensure the final version is clean.
>
>
>
> > **`Weakness 10`**: Section 7.1: The notations of metrics used should be explained in the main paper.
>
> `Response`: Thank you for this valuable suggestion. Following your advice, **we have moved the complete explanation for all evaluation metrics and their notations from Appendix C.1 into Section 7.1 of the revised manuscript**. All notations of metrics used in Table 1 are now clearly defined in the main manuscript before the results are presented.

---

> ### Author Response · Authors · 2025-11-19
> **(6/7) Responses by Authors**
>
> > **`Question 1`**: Could the authors elaborate how ToM is explicitly used? According to the reviewer's understanding, such details are rather limited in the main paper.
>
> `Response`: ToM is explicitly implemented in our framework through the **Hierarchical Reviewer Profile Modeling (Section 4.1)**. This module functions as the cognitive core of our pipeline, and it is explicitly used in two distinct phases: **Mental State Inference and State-Conditioned Generation**.
>
> ### 1. **Mental State Inference (The "Mind")**
> Rather than simply summarizing the review text, this module explicitly models the reviewer's latent mental state. (1) **At the Macro-level**, it acts as a profiler, inferring the reviewer's hidden attributes: their Overall Stance (e.g., Reject vs. Accept), Attitude (e.g., Skeptical vs. Constructive), and Expertise Level (e.g., Domain Expert vs. Generalist). (2) **At the Micro-level**, it performs "cognitive deconstruction"  on specific comments to identify the underlying intent—distinguishing, for instance, between a "fatal flaw criticism" (Methodology) and a "request for clarification" (Presentation).
>
> ### 2. **State-Conditioned Generation (The "Use")**
> it serves as a dynamic condition that explicitly dictates the downstream Strategy (Section 4.2) and Response (Section 4.3). (1) **From Mind to Strategy**: The generation of the rebuttal plan is mathematically conditioned on the ToM profile. For example, if the ToM module detects a "Skeptical Domain Expert" concerned with "Rigor", the model is directed to generate a strategy focused on evidence and technical depth. Conversely, if the profile indicates a "Confused Generalist", the strategy shifts towards clarification and pedagogical reframing. (2) **From Strategy to Response**: Finally, the text generation module uses this strategic blueprint to synthesize the response, ensuring the tone and content align with the inferred reviewer persona.
>
> To clearly illustrate this flow, we provide a concrete example from our results:
>
> | Comment   |  In the introduction and method section, it's hard for me to follow the motivation why it is beneficial to think of instant-NGP and tensorRT in the framework of gauge transform. This is not clear to me—what's the theoretical benefit (cleaner? easier to reason about different mappings?) or the empirical advantages. I would also encourage the authors to include a background section about gauge transform as it can benefit readers without enough background. |
> |----------------|----------|
> | RebuttalAgent: | Generated |
> | ToM Analysis       |  **Macro Profile**: {'overall_stance': 'Probably Accept', 'overall_attitude': 'Constructive', 'dominant_concern': 'Experimental Rigor', 'reviewer_expertise': 'Domain Expert', 'confidence': 10}. **Micro Intent**:{'category': 'Presentation & Clarity', 'sub_category': 'Lack of Detail', 'severity': 'Major', 'confidence': 10} |
> | Strategy      | **1**. Explicitly articulate the theoretical benefit: Clearly state that the gauge transform framework provides a principled, mathematically unified language for understanding and generalizing diverse coordinate-based representations, enabling cleaner, more transparent reasoning about mappings and invariances. **2**. Illustrate with concrete examples: Provide explicit, step-by-step examples showing how a single gauge transformation can unify instant-NGP, tensorRF, and triplanes, and how this unification simplifies analysis and design. **3**. Commit to improved exposition: Add a dedicated background section on gauge transformations, clearly defining the mathematical framework and illustrating its relevance to the specific methods discussed, and explicitly state these revisions in the response. |
> | Response     |  Thank you very much for this important suggestion. We agree that the motivation for framing instant-NGP and tensorRF within the language of gauge transformations was not sufficiently explicit in the original manuscript. We will substantially expand the introduction and method sections to clarify both the theoretical and practical benefits of this approach. Theoretically, the gauge transform framework provides a principled and mathematically unified language for describing and analyzing diverse coordinate-based representations, such as instant-NGP, tensorRF, and triplanes. By treating these models as different instances of the same underlying gauge structure, we can cleanly express their relationships, identify shared invariances, and generalize across them. For example, instant-NGP can be viewed as a gauge transformation of a standard MLP, tensorRF as a tensor-valued gauge, and triplanes as a planar gauge, each step made explicit through the mathematical structure of gauge transformations. This perspective allows for a more transparent and systematic reasoning process, enabling the design of new, hybrid, or adaptive coordinate representations that leverage the strengths of multiple paradigms ... |

---

> ### Author Response · Authors · 2025-11-19
> **(7/7) Responses by Authors**
>
> > **`Question 2`**: When evaluating the responses in the main experiment, is the score an average of comment-wise responses or considers the quality of the concatenated responses against each complete review?
>
> `Response`:  Thank you for this important question. **Our evaluation is performed on a comment-wise basis. The "Avg" scores reported in our main experiment are the average of all these individual comment-wise scores from the test set**. Specifically, our evaluation process is as follows: (1) Our framework is designed to address one "target comment" ($c_{target}$) at a time, as defined in our task formulation (Section 2). (2) Therefore, our evaluators (both the Rebuttal-RM and the human experts) assess the quality of a single response generated for a single corresponding comment. (3) The "Avg" score in Table 2 represents the mean of all scores obtained on all comment-response pairs.
>
> Thank you for your feedback. We will explain the evaluation process more clearly in the revised manuscript.

---

> ### Author Response · Authors · 2025-11-26
>
> Dear Reviewer snQL,
>
> Thank you once again for your thoughtful feedback on our paper. We appreciate the time and effort you have invested in reviewing our work.
>
> As we are nearing the end of the rebuttal period, we wanted to kindly follow up to ensure that you’ve had a chance to review our response to your comments. We hope that our rebuttal has addressed your concerns satisfactorily. If there are any additional points or further clarifications required, we would be happy to provide them promptly.
>
> Additionally, if our response has sufficiently addressed your concerns, we would greatly appreciate if you could consider reflecting this in your evaluation, including revisiting your score if appropriate.
>
> Thank you for your time and understanding. We greatly value your feedback and look forward to hearing from you.
>
> Best regards,
>
> Authors of Paper Submission 8600

---

> ### Author Response · Authors · 2025-11-28
>
> Dear Reviewer snQL,
>
> Thank you for your thoughtful and constructive feedback on our paper. We have carefully addressed your comments and suggestions, which have significantly helped us enhance the rigor and clarity of our work. We also appreciate the generally positive feedback regarding our work.
>
> **We have uploaded a revised version of the manuscript** where major changes are highlighted in blue. In the revised manuscript, we expanded our evaluation by updating Table 2 with new baselines and ablation studies to validate model robustness. We also clarified methodological details and moved key details into the main text for better readability.
>
> We believe these revisions fully address the questions raised. If there are any further questions or if additional clarification is needed, we would be more than happy to answer them.
>
> Best regards,
>
> Authors of Paper Submission 8600

---

### Author Response · Authors · 2025-11-28

Dear SAC, AC, and Reviewers,


Thank you very much for your time and helpful feedback. Your suggestions have helped us improve the clarity and quality of our paper. **We have uploaded a revised version of the manuscript with the main changes highlighted in blue**.

we have made the following specific changes:

- We added a detailed discussion and a comparison table. This clearly shows how our method is different from other agent-based approaches. We also added new baselines to better show the value of our framework.

- To show that our data preparation is reliable, we moved the validation details from the appendix to Section 3.1. We also moved the definitions of the evaluation metrics to Section 7.1 to make the paper easier to read.

- We added a more complete review of Theory of Mind in Section 4.1.

- We added new tests to compare our self-reward mechanism against external reward models (updated Table 2). This confirms that our method works well and is robust.

We believe these revisions fully address your questions. Please do not hesitate to reach out with any remaining concerns or queries. We are committed to ensuring all aspects of our submission are adequately addressed.

Many thanks!

Authors of Paper Submission 8600

---

### Author Response · Authors · 2025-12-01
**Rebuttal Summary and Major Updates of Our Manuscript**

Dear Program Chairs, Senior Area Chairs, Area Chairs and Reviewers:

We would like to thank the reviewers for the time and effort dedicated to evaluating our submission. It is encouraging that all reviewers recognized the value and novelty of our work.

Specifically, we would like to highlight the following strengths acknowledged by the reviewers:

1. `Novelty and Significance of the Perspective (Theory of Mind)`: **Reviewer LugC** highlighted the work's *"innovative perspective by framing academic rebuttal as a ToM reasoning task"*. **Reviewer hzng** found the motivation *"intuitive and reasonable"* and this direction *"worth exploring,"* and **Reviewer d1nh** described the task as *"very interesting"* and an *"important step for LLMs toward academic reasoning."*
2. `Comprehensive Framework and Resource Contribution (RebuttalBench & Pipeline)`: **Reviewer d1nh** commended the study for providing *"valuable resources,"* specifically citing the *RebuttalBench* benchmark and *RebuttalJudge* model, and offering a *"complete framework"* for development. **Reviewer LugC** praised the methodology, noting that the integration of SFT and reinforcement learning creates a self-improving training pipeline. **Reviewer hzng** specifically highlighted the *"useful finding"* that the self-generated reward mechanism effectively improves quality without relying on explicitly trained reward models.
3. `Extensive Experiments and Strong Performance`: **Reviewer snQL** emphasized that the improvement over baselines is *"significant."* **Reviewer hzng** and **LugC** both acknowledged that *RebuttalAgent* achieves *"better performance"* and *"consistent improvements"* across both automated and human evaluations. **Reviewer d1nh** further validated that the *"comprehensive experiments"* clearly demonstrate the agent's effectiveness.

### Summary of Major Updates
In response to the constructive feedback, we have significantly revised the paper. The major updates include:

1. **Extended Comparison with New Baselines**: We introduced powerful new baselines: **TSR-o3** and **TSR-GPT-4.1** (applying our ToM-Strategy-Response framework to o3 and GPT-4). Our 8B RebuttalAgent (**9.42 Avg**) consistently outperforms these enhanced advanced models, demonstrating that our framework contributes significant strategic capabilities beyond general reasoning improvements (Addressing **Reviewer d1nh** and **snQL**).

2. **Further Validation of Self-Reward Mechanism**: We conducted rigorous ablation studies comparing our self-reward mechanism against agents trained with external reward models (including **Rebuttal-RM and GPT-4.1**). The results confirm that external rewards lead to policy collapse and overfitting (high proxy scores but low human preference scores), whereas our self-reward paradigm achieves superior alignment with human experts (**9.57**), validating its robustness (Addressing **Reviewer d1nh** and **LugC**).

3. **Extension to Multi-Turn Discussions**: We constructed a new **Multi-Turn Dialogue (MTD-200)** dataset consisting of 200 real multi-turn dialogues. Blind human evaluations indicate that RebuttalAgent excels in dynamic interactions, improving **Constructiveness** in initial responses and **Persuasiveness** in follow-up rounds, proving its applicability to real-world, iterative academic communication (Addressing **Reviewer hzng**).

4. **Generalizability across Model Architectures and Scales**: To verify that our method is not backbone-specific, we extended our evaluation to smaller and diverse architectures, specifically **Qwen3-4B** and **Llama-3.1-8B**. The significant performance gains observed across these varying sizes (e.g., boosting the average score of Qwen3-4B from **7.69** to **8.98**) confirm that the our TSR framework and self-reward mechanism are transferable and effective (Addressing **Reviewer LugC**).

5. **Clarifications and Presentation Enhancements**: We have refined the manuscript. Key updates include moving the **"LLM-as-Extractor" validation** (demonstrating 98% accuracy) and formal **metric definitions** from the appendix to the main text for readable. Additionally, we expanded the **Theory of Mind (ToM) background** in Section 4.1 and corrected typographical errors (Addressing **Reviewer snQL** and **hzng**).

We have addressed all feedback and conducted extensive new experiments to resolve the reviewers' concerns. We are particularly encouraged that **Reviewer LugC has explicitly acknowledged the effectiveness of our response** and maintained a positive evaluation. Our new empirical evidence firmly establishes the robustness of our framework.

We deeply appreciate the time and effort provided by the Area Chair, Senior Area Chair, and Program Chairs during this process. We sincerely hope that the concrete improvements made during the rebuttal will weigh positively in your final assessment. Thank you all for your dedication to the review process.

Best regards,

Authors of Paper Submission 8600

---

### Meta-Review · Area_Chair_AvnA · 2026-01-07

**Summary:**

The paper contributes (1) a Theory of Mind (ToM)-inspired reasoning framework for rebuttal generation, (2) a curated benchmark (RebuttalBench) for instruction tuning in the rebuttal setting, and (3) an automatic evaluation model (Rebuttal-RM) to evaluate rebuttal quality.

Reviewers broadly agree that this is an interesting and meaningful research direction, representing a step toward more complex academic reasoning in LLMs. The released resources, particularly the RebuttalBench and the Rebuttal-RM, are valuable contributions that could benefit future research.

While reviewers found the direction compelling, they noted some concerns. The framework mainly combines existing techniques, baselines could be stronger, and the approach focuses on writing only rather than broader rebuttal reasoning.

Overall, the reviewers generally agree that the paper tackles an important and underexplored problem and provides meaningful resources and insights. The strengths slightly outweigh the outstanding issues.  I therefore recommend acceptance.

I strongly encourage the authors to incorporate the new experiments and discussions from the rebuttal into the revised paper.

**Reviewer Concerns:**

- For reviewers snQL and LugC, most of the concerns are addressed.

- Reviewer d1nh may still have concerns about the limited technical novelty and remains unconvinced that the baseline comparisons are sufficiently strong. In particular, they suggest including powerful non-trained baselines (e.g., o3 or GPT-4 variants) evaluated directly under the proposed ToM framework. While the rebuttal added comparisons with o3+TSR and GPT-4.1+TSR, the results appear mixed rather than clearly dominant, making claims of consistent superiority less conclusive.

- Reviewer hzng may still be concerned that the current approach focuses primarily on rebuttal writing and does not address broader rebuttal reasoning, such as proposing additional experiments. Experiments demonstrating this capability would strengthen the paper.

**Reviewer Scores:**

- Reviewer snQL: Score increased from 4 -> 6; concerns regarding originality, quality, and clarity were resolved.

- Reviewer d1nh: Score remains 6; most of the concerns are addressed.

- Reviewer LugC: Score remains 6; the reviewer indicated they would keep the score.

- Reviewer hzng: Score remains 4; concerns persist regarding the limited scope of rebuttal reasoning beyond writing.

---

### Decision · Program_Chairs · 2026-01-26

Accept (Poster)